# Local Kernel Ridge Regression
# for Scalable, Interpolating, Continuous Regression

**Mingxuan Han**                                                                *u1209601@utah.edu*
*School of Computing*
*University of Utah*

**Chenglong Ye**                                                              *chenglong.ye@uky.edu*
*Dr. Bing Zhang Department of Statistics*
*University of Kentucky*

**Jeff M. Phillips**                                                              *jeffp@cs.utah.edu*
*School of Computing*
*University of Utah*

**Reviewed on OpenReview:** *https://openreview.net/forum?id=EDAk6F8yMM*

## Abstract

We study a localized version of kernel ridge regression that can continuously, smoothly interpolate the underlying function values which are highly non-linear with observed data points. This new method can deal with the data of which (a) local density is highly uneven and (b) the function values change dramatically in certain small but unknown regions. By introducing a new rank-based interpolation scheme, which can be interpreted as a variable bandwidth Nadaraya-Watson Kernel Regression, the interpolated values provided by our local method can be proven to continuously vary with query points. Our method is scalable by avoiding the full matrix inverse, compared with traditional kernel ridge regression.

## 1 Introduction

For decades the idea of interpolating data has been scorned in machine learning, since it was prone to overfit, and then not generalize to new data. However in the last decade, models with significantly more parameters than data points have become the norm, and these models achieve the best generalization when they have little or no error on the training data. This suggests we should revisit models that can be designed to explicitly interpolate the data.

In this paper we study (mostly low-dimensional) regression problems, but with somewhat non-standard assumptions. First, we assume the noise is very small so the model should more-or-less interpolate the training data. Second, we assume individual data points may be important for the model in a local sense. The data may be strategically placed either due to generation from a physical simulation, or because it was chosen adaptively to fill in areas that lacked coverage. Third, we assume that there is a smooth or continuous underlying model behind the data, e.g., driven by physics or some natural phenomenon. However, the underlying model itself can be complicated with many local features at various scales, which makes the exact recovery from random sampling challenging. Hence the estimate should be smooth, and at least continuous. Fourth, the data size may be reasonably large so (a) a full matrix inverse may be impractical in building the model, and (b) the model evaluation should be much faster than linear in the data size.

One approach towards dealing with these challenges is to keep the estimated model "local" in that for any query $q \in \mathbb{R}^d$ the estimated model only depends on nearby data points. Evaluation time should only depend on these local points or simple parameters. Starting from this possible vantage, we propose a *localized kernel*

*ridge regression* (local KRR) which adjusts the bandwidth/scale locally, is very scalable, and moreover is continuous while also nearly interpolating the data.

One motivation for this modeling and new algorithm is the observation (Belkin et al., 2018b; 2019a) that many modern large-scale machine learning models achieve 0 or nearly-0 error on training data. This has led to the study of simpler models where interpolation can be achieved (Belkin et al., 2019b;a; Hastie et al., 2020; Belkin et al., 2018b; Liang & Rakhlin, 2020). A second motivation arises in modeling physics simulations (Pope, 2013; Van Oijen & De Goey, 2000) where data is generated according to a well-behaved and constrained pattern, and with very little error compared to the traditional complexity of the model and sampling of the data. Such data may be generated by simulating non-linear PDE systems with uneven local density as more data points are generated and more detail is required where the PDE evolves slowly, and fewer data points and less detail where it evolves rapidly. Models with a global scale parameter or traditional error measures (e.g., RMSE) can miss important features. Both settings have a scale where the cost of model learning and evaluation is non-trivial.

To deal with challenges posed in such datasets, our local KRR builds (nearly) interpolating regression models centered around a set of model points $M$ chosen to adapt to data density. Using a fast k-nearest neighbor (KNN) search, we determine which local data $X$ near each $m_i \in M$ to be included in building a local model. At any query point $q$, we use a weighted average of nearby pre-built models, as determined by their model points. We introduce a new rank-based interpolation scheme to ensure that this weighted average continuously varies with the query points, and hence the full model is continuous. As elaborated on in Section 4.1, this can be interpreted as a variable-bandwidth Nadaraya-Watson kernel regression estimator over the local models. Our analysis guarantees the continuity of such an estimator. This avoids either a full KNN search on $X$ at query time, avoids truncating a continuous kernel (causing discontinuity), and avoids building a simplicial complex to allow for consistent barycentric interpolation. Local KRR is a locally-defined, nearly-interpolating model where the most expensive part of evaluation is a KNN query (for small $k$) on a reduced set of model points $M \subset X$. It is non-parametric and locally-optimized, yet efficient and scalable. We empirically demonstrate interpolation ability, generalization accuracy, and efficiency of this new local KRR model by comparing with global models, discontinuous local models, and non-learned interpolating local models. For completeness, we also identify some standard data assumptions under which we can guarantee our model still achieves convergence in the standard global $\ell_2$ sense.

## 2 Background and Related Work

**Kernel Ridge Regression estimator.** Let $(X, y) \subset \mathbb{R}^d \times \mathbb{R}$ be a set of $n$ observed data points with explanatory variables in $X$ and response variable in $y$. We will leverage a positive definite kernel $\mathsf{k} : \mathbb{R}^d \times \mathbb{R}^d \to \mathbb{R}$, and will by default use the un-normalized Gaussian kernel $\mathsf{k}(x, q) = \exp(-\|x - q\|^2/b^2)$ for a bandwidth parameter $b > 0$. This gives rise to a kernel matrix $\mathsf{K} \in \mathbb{R}^{n \times n}$ where $\mathsf{K}_{i,j} = \mathsf{k}(x_i, x_j)$ for $x_i, x_j \in X$. The kernel regression estimator at a point $q \in \mathbb{R}^d$ has the form: $\mu(q) = \sum_{i=1}^n \alpha_i \mathsf{k}(x_i, q)$, where $\alpha$ is known as dual coefficients of KRR, e.g. in Murphy (2012). The kernel ridge regression (KRR) estimator augments the standard least squares formulation with a ridge term to regularize towards a simpler model with a robust solution. Specifically, given a ridge parameter $\eta$ its goal is to minimize $\sum_{i=1}^n (\mu(x_i) - y_i)^2 + \eta \alpha^T K \alpha$, for which the optimal $\alpha$ can be solved as $\alpha = (\mathsf{K} + \eta I_n)^{-1} y$.

Extensive work has shown the generalization properties of and risk bounds for KRR (c.f. Steinwart et al. (2009); Eberts & Steinwart (2013); Cui et al. (2021)). In an idea related to our proposed local KRR extension, Zhang et al. (2013) designed a scalable variant of KRR that splits the data into parts and takes their average. The main distinctions with our local KRR approach are that Zhang *et.al.*'s model (a) splits data randomly, not defined by local structure, (b) enforces a global bandwidth parameter so cannot adjust to local variation in data density and model complexity, (c) does not aim to or achieve interpolation.

Other work, knn-svm Hable (2013) and local-svm Meister & Steinwart (2016), also considered kernel regression models where the data is split to form many smaller models, and these works defined the splits into spatially contiguous regions – as does our method. However these (a) local-svm combines local models into a full one by selecting a nearest one (different from our weighting scheme) and does not ensure global continuity since the models do not match at the boundary between nearest models. For knn-svm, the model is defined

implicitly, and built at query time on the $k$ nearest points – so the issue persists. (b) Neither aims for, or analyzes, the (near-)interpolation properties. (c) local-svm defines local regions by volume, and hence do not refine subject to local data density.

**Interpolation Learning.** Recent work (Belkin et al., 2018b; Bartlett et al., 2020; Liang & Rakhlin, 2020) has observed that state-of-the-art methods for large complex models (e.g., deep learning, those exhibiting double descent (Belkin et al., 2019a)) which have more parameters than data tend to perform best on out-of-sample data. Belkin et al. (2018b) established a theoretical foundation of generalization error analysis of such phenomenon based on the local interpolation scheme by using an unusual kernel called a singular kernel Shepard (1968). Follow on work (Belkin et al., 2019b) showed that Nadaraya-Watson kernel regression (NWKR, a form of kernel regression that does not "learn" the $\alpha$ parameters) with a singular kernel has a type of guarantee of statistical optimality. Liang & Rakhlin (2020) explained how using a nonlinear kernel, in the high-dimensional setting (e.g., $d > n$), and with the ridge parameter $\eta$ goes to zero, global KRR can achieve zero training error and generalize well on new data; see also related results of Liu et al. (2021); Elkhalil et al. (2020); Karoui (2010). However, we are unaware of comparable work about generalization for KRR with ridge parameter $\eta = 0$ or small, and in low or moderate dimensions ($n > d$).

Belkin et al. (2018b) identified another method to learn an interpolating model to $(X, y)$ without using kernels. This method builds a simplicial complex on $X$ and simply interpolates the values of each $y_i$ within each simplex using $x_i$ as a vertex (e.g., using barycentric coordinates). They show for data generated by smooth manifolds, this will converge with enough samples. However, the size of a simplicial complex (combinatorially describing each simplex) unfortunately grows exponentially in the dimension $d$. While our work is inspired by this, it devises a new interpolation scheme that does not rely on the simplicial complex – it only uses (fast) nearest neighbor searching. We also show similar convergence analysis is amenable to our method.

**Fast Nearest Neighbor Search.** The main computation cost of our algorithm will be invoking KNN search and its relatives. This is an area of computer science that has seen enormous recent progress (Arya & Mount, 1993; Andoni et al., 2015; Li et al., 2019; Ram & Sinha, 2019; Bernhardsson, 2021; Johnson et al., 2021), and is nearing maturity in theory and practice. There are several related operations such libraries can perform, which we will employ in our algorithms: $p$-NN$_X(q)$ returns the $p$ nearest neighbors of $q$ in $X$; $p$-NN$_X(q)$ returns only the $p$th nearest neighbor of $q$ in $X$; and RANGE$_X(q, r)$ returns all points in $X$ within a distance $r$ from $q$. For instance, on 100-NN$_X(q)$ queries (e.g., on SIFT1M (Jégou et al., 2011) with $n = 10^6$, $d = 128$), libraries can return beyond 1000 queries a second (Bernhardsson, 2021). We employ a state-of-the-art method FAISS (Johnson et al., 2021).

## 3 Local Kernel Ridge Regression

Consider again the dataset $X \subset \mathbb{R}^d$. Instead of building one global KRR model over the $X$, we propose a data density-adaptive localized version of KRR.

### 3.1 Constructing Local KRR

To construct the local KRR model, we first need to determine a set of model points $M \subset X$. Then build a model $\mu_j$ around each model point $m_j \in M$.

**Determine $M$.** We employ the following method to choose the set $M$. It is iterative, and starts with an arbitrary point $m_1 = x_i \in X$ as the center of the first region. It then sets the radius of $m_1$ so that it contains exactly $\ell$ points. We maintain for each point $x_i \in X$ by how many regions it has been covered. In each iteration, we choose an arbitrary (this is done at random) point that has been covered fewer than $t$ times. In particular, if there are some points which have never been covered, we choose among these first. If all points have been covered at least $j$ ($j < t$) times but some only $j + 1$ times (again $j + 1 < t$), then we choose a new center among those covered only $j + 1$ times (not ones covered $j + 2$ times). This is repeated until the coverage of each point is at least $t$ times.

**Learning the local models.** When building the local KRR, for each model point $m_i \in M$, we determine $\ell\text{-NN}_X(m_i)$ and $b_i = \|m_i - \ell\text{-NN}_X(m_i)\|$. The $b_i$ becomes the bandwidth for the kernel k used in KRR, following similar approaches for variable bandwidth selection (Loftsgaarden & Quesenberry, 1965; Terrell & Scott, 1992; Fan & Gijbels, 1992). Next we retrieve the nearby points on which to build our model $X_i = \text{RANGE}_X(m_i, \lambda \cdot b_i)$. Here $\lambda > 1$ is an expansion factor; by default $\lambda = 3$; a common choice of where to truncate a kernel (Robert, 1995). It ensures the points used to build each model sufficiently cover the neighborhood beyond 1 bandwidth, and also ensures the models provide sufficiently overlapping coverage of $X$ for guarantees on evaluation. In practice, we keep $\lambda$ small so that the models are constant size and run efficiently.

Now we can construct our local KRR model $\mu_i$, so it evaluates a query point $q \in \mathbb{R}^d$ as $\mu_i(q) = \sum_{x_j \in X_i} \alpha_j \mathsf{k}(x_j, q)$, where $\alpha = (\mathsf{K}_i + \eta I)^{-1} y^{(i)}$, $\mathsf{K}_i$ is the kernel matrix $\mathsf{K}$ restricted to $X_i$, and $y^{(i)} \in \mathbb{R}^{|X_i|}$ is the restriction of $y$ to points in $X_i$. We summarize the training of the local KRR model in Algorithm 1.

---

**Algorithm 1** LOCAL KRR TRAINING

---

1: Determine $M$.
2: **for** $m_i \in M$ **do**
3:     Get $\ell\text{-NN}_X(m_i)$; set $b_i = \|m_i - \ell\text{-NN}_X(m_i)\|$.
4:     Retrieve $X_j = \text{RANGE}_X(m_i, \lambda b_i)$
5:     Learn KRR model $\mu_i(\cdot)$ on $(X_i, y^{(i)})$ with $b_i$.
6: **end for**

---

### 3.2 Evaluating Local KRR

To evaluate at a query point $q \in \mathbb{R}^d$, instead of using the single evaluation function $\mu_i(\cdot)$, we use a weighted average evaluation based on the $k$ closest model points $M_q \subset M$. This only needs a fast $M_q = k\text{-NN}_M(q)$ call over the model points and the rest is pre-trained. Let $t \geq k \geq d+2$, as we will see in Section 4.1, this will allow us to provide results on continuity of the local KRR model. We generally use $t = k = d + 3$ to be conservative. The evaluation of a query point $q$ is then a weighted average over these $k$ models $\mu(q) = \sum_{m_i \in M_q} p_i \mu_i(q)$ with the proportion values $p_i \in (0, 1]$ and $\sum p_i = 1$ explained next.

**KNN Model Interpolation.** Given a query point $q \in \mathbb{R}^d$, let $M_q = m_1, m_2, \ldots, m_k \subset M$ be the $k$-nearest neighbors to $q$ in $M$. Set $r_j = \|q - m_j\|$. Adopt the convention that $r_j \leq r_{j+1}$. Assign weights to each of the points in $M_q$ as $w_j = (r_k - r_j)/(r_k - r_1)$. Observe that this ensures $w_k = 0$ and $w_1 = 1$. Finally, we assign each model $\mu_j$ a proportion $p_j = w_j/(\sum_{i=1}^k w_i)$. We summarize how to evaluate $\mu(q)$ using KNN model interpolation in Algorithm 2.

---

**Algorithm 2** LOCAL KRR EVALUATION AT $q$

---

1: Get $M_q = \{m_1, m_2, \cdots m_k\} = k\text{-NN}_M(q)$
2: Let $r_j = \|q - m_j\|$ for $j \in [1...k]$
3: Set weight $w_j = \frac{r_k - r_j}{r_k - r_1}$ for $j \in [1...k]$
4: Return the weighted average evaluation of $q$ as: $\mu(q) = \frac{\sum_{j=1}^k w_j \cdot \mu_j(q)}{\sum_{j=1}^k w_j}$

---

## 4 Properties of Local KRR

The desired properties for our regression model $\mu$ are:

1. it adapts to the local density
2. it is efficient to evaluate near data $X$
3. it is continuous
4. it (nearly) interpolates data points

The adaption to the local density follows by how we adaptively choose model points $M$, set the bandwidth, and will be demonstrated empirically in Section 5.

The efficiency is due to each model being built on a constant number of points (determined by constant parameters $\ell$ and $\lambda$). After selecting model points, each training requires a fast $\text{RANGE}_X$ call, and is then constant time training. Similarly, the evaluation requires a fast $k$-$\text{NN}_M$ call, and then evaluating the $k$ pre-trained models, which each require a constant (depending on $\ell$ and $\lambda$) amount of operations. We demonstrate its empirical scalability in Section 5.

We show the continuity and the (near)-interpolation properties in the following two subsections.

### 4.1 Continuity of Local KRR

Some technical calculation-focused proofs are deferred to Appendix A.

**Interpretation as NWKR estimator.** We first note that weighted interpolation of the local models can be interpreted as a variable-bandwidth Nadaraya-Watson kernel regression (NWKR) estimator with a triangle kernel $K(q, m) = \max\{0, 1 - \|m - q\|/h\}$ with bandwidth $h = r_k$ chosen as the distance to the $k$th nearest neighbor model. Under NWKR, let $W = \sum_{m_i \in M} K(q, m_i)$ and the estimator at $q$ is $\frac{1}{W} \sum_{m_i \in M} w_i \mu_i(q)$. With this triangle kernel, only the nearest $k - 1$ models have non-zero weights, which are $K(q, m_i) = 1 - r_i/r_k$ and (via multiplying by $r_k/(r_k - r_1)$) are proportional to the weights considered in our interpolation $w_i = \frac{r_k - r_i}{r_k - r_1}$. While the continuity of fixed-bandwidth NWKR is known (Ferraty et al., 2011), the following analysis can be interpreted as analyzing the continuity for this variable bandwidth condition under the triangle kernel.

**Continuity of KNN Interpolation.** The first step towards formalizing the continuity of local KRR is to understand the continuity of the KNN interpolation scheme. Consider continuously moving the position of a query point $q$. As any model point moves into the set of $k$ nearest of $q$, its weight is initially zero. So there is no boundary effect discontinuity in the contribution of the nearest $k$ models when suddenly a different set of points are the $k$ nearest. Also, when two points change their relative position in the sorted order, they have the same weight. So again there is no discontinuity in the weight as the relative order of model points change within that nearest $k$.

To formalize that KNN interpolation proportions (the $p_j$ values) are continuous in choice of $q$, we only need to assume that $M$ is in *general position* (Todd, 1969); that is, no $d + 2$ points are all equidistant from any query point. We prove a Lipschitz bound with respect to the movement of $q$, and it will depend on a quantity $\bar{R}_k = \frac{1}{k} \sum_{i=1}^{k}(r_i - r_1)$. Recall, that $r_j$ is the distance from $q$ to the $j$th closest model point. For $k \geq d + 2$, and $M$ in general position, then $\bar{R}_k > 0$.

The Lipschitz factor will be smaller when $(r_k - r_1)$ and $\bar{R}_k$ is large. The same is true for a simplicial interpolation based on the standard Delaunay complex Aurenhammer et al. (2013). It is common when dealing with that complex to assume general position, since it is not well-defined otherwise.

Let $R_k = \sum_{i=1}^{k}(r_k - r_i)$. Recall the proportion $p_j$ for model $\mu_j$ in the weighted average is normalized by the total weight, and is written

$$p_j = \frac{r_k - r_j}{r_k - r_1} \frac{(r_k - r_1)}{\sum_{i=1}^{k}(r_k - r_i)} = \frac{r_k - r_j}{R_k}.$$

Now we need to consider moving the query point $q \in \mathbb{R}^d$ by some small amount $\delta \in \mathbb{R}$. To formalize this consider some other point $q' = q + \delta u$, where $u \in \mathbb{R}^d$ and $\|u\| = 1$. So the perturbation is a change in $\mathbb{R}^d$, but we will only care about the magnitude $\delta$.

**Lemma 4.1.** *Consider moving the query $q$ a distance at most $\delta \leq \frac{\bar{R}_k}{4}$. The proportion of model $\mu$ comprised of model $\mu_j$, denoted $p_j$, changes by at most $4\delta/\bar{R}_k$.*

Now make a stronger assumption on $M$, so it is evenly distributed. Its density may change, but it should do so in some local way. For $\gamma \geq 0$ we say $M$ is *($\gamma, k$)-distributed* if any point $q \in \mathbb{R}^d$ satisfies $\bar{R}_k/r_k > \gamma$. If a point set is $(0, d + 2)$-distributed, it is degenerate in the sense that $d + 2$ points are equidistance to some

point $q$, and so not in general position. For $k \geq d + 2$ we can expect $\gamma > 0$. We can now understand the continuity bound Lemma 4.1 in terms of $\gamma$.

**Corollary 4.2.** *For model points $M$ that are $(\gamma, k)$-distributed, consider moving the query $q$ a distance at most $\delta \leq \bar{R}_k/4$. The proportion of model $\mu$ comprised of model $\mu_j$, denoted $p_j$, changes by at most $\frac{\delta}{r_k}\frac{4}{\gamma}$*

*Proof.* Recall that this implies $\frac{\bar{R}_k}{r_k} \geq \gamma$ if $M$ is $(\gamma, k)$-distributed. Thus $\bar{R}_k \geq \gamma r_k$, and $\delta/\bar{R}_k \leq \delta/(r_k\gamma)$. □

Let us interpret this bound. As a point $q$ moves a distance $\delta$, then the proportion that a model can change moves at most proportional to $\delta$; it has a Lipschitz parameter $\frac{4}{r_k\gamma}$. This parameter has two parts.

This first is $r_k$; this has the same unit and scaling as the data. All Lipschitz bounds that convert between two things with different units (e.g., $q$ and $p_j$) need some such term to make sense. In this case, however, we note that $r_k$ is *local*. That is, if the local density of points changes, then the change in effect from $q$ on $\mu_j$ also changes. Note that using the more generic form in Lemma 4.1 had a similar effect.

The second part is $\gamma$, which is a notion of how close a point set is to degeneracy, measured in a density agnostic way. This value has no units, so it does not interfere with the properties discussed about $r_k$. Now recall that as the point set becomes more degenerate, then $\gamma$ goes towards 0. That makes the stability worse; a smaller change in $q$ can effect a larger change in $p_j$. So less degenerate distributions $M$ have more stable relationship between the movement of the query, and the change in the functions used.

**Local KRR model stability.** Now based on the Lemma 4.1 and Lipschitz continuity of each $\mu_j$ we can show that the model $\mu$ is Lipschitz continuous.

**Theorem 4.3.** *Assume $M$ is $(\gamma, k)$ distributed, for any $\mu_j \in M$, for $q, q' \in \mathbb{R}^d$ with $\|q - q'\| = \delta \leq \bar{R}_k/4$ and $|\mu_j(q') - \mu_j(q)| \leq L \cdot \delta$ and $\mu_j(q) \in (-T, T)$, then $|\mu(q') - \mu(q)| \leq \frac{4\delta(k-1)T}{\bar{R}_k} + L\delta$.*

From the Theorem 4.3 we can see that our model $\mu$ is Lipschitz continuous bounded by a constant depending on the Lipschitz constant of each $\mu_j$ and also depend on the range of each $\mu_j$. And a bounded range of $\mu_j$ in $[-T, T]$ is common for kernel ridge regression (Steinwart et al., 2009; Eberts & Steinwart, 2013; Meister & Steinwart, 2016). The $(k - 1)$ shown in the bound is typically $O(d)$, e.g. $k = d + 3$.

Let's compare this to the simplicial-barycentric interpolation (Belkin et al., 2018b). Both require general position, although ours can loosen this requirement by increasing $k$. Theirs relies on knowing the $(d+1)$-simplex (e.g, triangle in $\mathbb{R}^2$) that a query point lies in to determine the proportions. Our KNN interpolation does not, and only needs the $k$-nearest neighbors. As the complexity of the simplicial complex grows exponential in $d$, ours should be more generally applicable.

## 4.2 Local KRR (Nearly) Interpolates $(X, y)$

We now show how KRR and our proposed local KRR model nearly interpolates a data set $(X, y) \subset \mathbb{R}^d \to \mathbb{R}$. We say a function $g : \mathbb{R}^d \to \mathbb{R}$ $\psi$-*interpolates* $(X, y)$ if for all $(x_i, y) \in (X, y)$ we have that $|g(x_i) - y_i| \leq \psi$. While a kernel regression model can 0-interpolate any data set, it is unstable, and may result in a extreme model, e.g., with very large gradient. However, for a ridge model to nearly interpolate, we need to include some assumptions on the niceness of the data, otherwise, the ridge regularization will force the learned model to favor other simplicity factors over fitting the data.

**Small noise, good fit assumption.** We now describe a noise property for data $(X, y) \in \mathbb{R}^d \times \mathbb{R}$, for kernel ridge regression models that leads to bounds in interpolation. It involves parameters for noise $\nu$ and goodness of fit $A$. We say data that satisfy this have an $(\nu, A)$-*fit*. This assumes a known reproducing kernel (e.g., a Gaussian) with a known bandwidth $b$.

Specifically, $(\nu, A)$-fit data $(X, y)$ should be so that for each $y_i$ there is a $y_i'$ so $|y_i - y_i'| \leq \nu$ and so there exists a model $f_{\tilde{\alpha}}(q) = \sum_{x_j \in X} \tilde{\alpha}_j \mathsf{k}(x_j, q)$ so that $f_{\tilde{\alpha}}(x_i) = y_i'$ and so the the norm $\tilde{\alpha}^T \mathsf{K} \tilde{\alpha} \leq A$. We think of this $f_{\tilde{\alpha}}$ as a potential generating model.

For any reproducing kernel $\mathsf{k}$, and any data where $x_i \neq x_j$ for all $i \neq j$ and $y_i \in [-T, T]$ with $T$ bounded, there exists some interpolating solution. However, this solution is often unstable (involving $\mathsf{K}^{-1}$), and so a small ridge term $\eta$ is added, and solve for $\hat{\alpha}_\eta = \arg\min_\alpha \sum_i (y_i - f_\alpha(x_i))^2 + \eta \alpha^T \mathsf{K}\alpha$ instead. By duality, there exists some decreasing function of $\eta$ we call $h(\eta)$, so the ridge solution $\hat{\alpha}_\eta$ is the one that minimizes $\sum_i (y_i - f_\alpha(x_i))^2$ restricted to $\alpha^T \mathsf{K}\alpha \leq h(\eta)$. Hence, for any reproducing kernel $\mathsf{K}$, and data set $(X, y)$ it is $(\nu, A)$-fit for some parameters $\nu$ and $A$. The ones with very small $\nu$ and small $A$ are those where points with nearby $x_i$ values also have similar $y_i$ values.

As motivation for this model, we consider data generated from some real smooth phenomenon with high precision. So if $y \in [-T, T]$ we may expect $\nu/T < 10^{-6}$ is the amount of precision in the results. Similarly, if the data $X$ is well-spread and close explanatory values $(x_i)$ have similar response values $(y_i)$, the model will not have heavy reliance on individual data points, and hence the corresponding model coefficients $\tilde{\alpha}_j$ will be bounded. For instance, when $\mathsf{k}(x, x) = 1$, and $f_{\tilde{\alpha}}$ is built on a fairly uniform average over about $k$ elements, then $\tilde{\alpha}_j \approx y_j/k$ and $\tilde{\alpha}^T \mathsf{K}\tilde{\alpha}$ is roughly $T^2$.

**Analysis of the small noise, good fit setting.** Now we analyze the accuracy of the model $\hat{f} = f_{\hat{\alpha}}$ where $\hat{\alpha} = (\mathsf{K} + \eta I)^{-1} y$ is chosen to minimize the standard sum of squared errors on $(X, y)$ (the observed data, not on $(X, y')$) with a ridge penalty

$$\mathsf{Error}_\eta(\alpha) = \sum_{i=1}^{n} (y_i - f_\alpha(x_i))^2 + \eta \alpha^T \mathsf{K}\alpha.$$

For $(\nu, A)$-fit data, the potentially generating model $\tilde{\alpha}$ satisfies that $\mathsf{Error}_\eta(\tilde{\alpha}) \leq \sum_{i=1}^{n} \nu^2 + \eta A = n\nu^2 + \eta A$. Moreover $\mathsf{Error}_\eta(\hat{\alpha}) \leq \mathsf{Error}_\eta(\tilde{\alpha})$. To understand how well this interpolates, meaning how much deviation can occur on a single data point, in the worst case this sum of squared errors is due to a single data point $(x_i, y_i)$. We can summarize in the following lemma.

**Lemma 4.4.** *Consider an $(\nu, A)$-fit data set of size $n$. Then for the learned $\hat{\alpha} = (\mathsf{K} + \eta I)^{-1} y$ with corresponding model $f_{\hat{\alpha}}$, ensures that for each $(x_i, y_i)$ it nearly interpolates so $|y_i - f_{\hat{\alpha}}(x_i)| \leq \sqrt{n\nu^2 + \eta A}$.*

**Local Interpolation.** Next we adapt the $(\nu, A)$-fit assumption to a model that requires a local fit, with adaptive bandwidth. With high density in $X$, this allows to reduce the bandwidth and fit more variation in the $y_i$ values, relative to $x_i$ change. In sparse regions, larger bandwidths allow for smoother functions that generalizes over a greater span. The next definition is based on the common bandwidth selection method via the distance to the $k$th nearest neighbor.

We say a data set $(X, y)$ and associated reproducing kernel $\mathsf{k}$ is *locally $(\nu, A, \ell)$-fit* if for any point $x_i \in X$, we can define a bandwidth as $b_i = \|x_i - \ell\text{-NN}_X(x_i)\|$, so that the data set $X_i = \ell\text{-NN}_X(x_i)$ is $(\nu, A)$-fit.

This modeling almost directly applies to our proposed local KRR algorithm. Under the local $(\nu, A, \ell)$-fit assumption, each model $\mu_j$ constructed at a model point $m_j \in X$ nearly interpolates the data $X_j = \{x \in X \mid \|x - m_j\| \leq \lambda b_j\}$ it is built on, with error at most $\sqrt{n_j \nu^2 + \eta A}$ where $n_j = |X_j|$. To apply this to the full local KRR model, we need a property on the relationship between $M$ and $X$.

Consider two subsets $X, M \subset \mathbb{R}^d$ parameter $\ell, k > 0$ and a real value $\lambda > 0$. For any $x \in X$ let $M_{x,k} \subset M$ be the $k$ points in $M$ closest to $x$. And non-symmetrically, given any point $m \in M$ let $X_{m,\ell,\lambda} \subset X$ be the points in $X$ that are within a distance $\lambda b(\ell)$ from $m$, where $b(\ell) = \|m - \ell\text{-NN}_X(m)\|$. We say $X, M \subset \mathbb{R}^d$ are $(\lambda, \ell, k)$-*balanced* if for any point $x \in X$, for each $m \in M_{x,k}$ we have that $x$ is in $X_{m,\lambda,\ell}$. This captures that $X$ and $M$ have similar densities in a key way, and that their point sets do not vary in density too abruptly. The difference in $\ell \neq k$ allows for different sizes in $|X|$ and $|M|$. In principal the ratio $\ell/k$ increases with $|X|/|M|$ (recall $M \subset X$), but will also depend on the intrinsic dimensionality in some way. Hence, we just leave the two parameters separate. The factor $\lambda > 1$ (even after adjusting for size) accounts for some strange boundary conditions and imbalance between the data sets. We define this as a radius $\lambda b(k)$, and not the $(\lambda k)$th closest point since the use of $\lambda b(k)$ corresponds to how we build a model around each $m$ using a kernel with bandwidth defined as $b(k)$.

**Theorem 4.5.** *If the data $X$ and model points $M$ are $(\lambda, \ell, k)$-balanced, and locally $(\nu, A, k)$-fit, then $\mu$ built using local KRR with ridge parameter $\eta$, region expansion $\lambda$, and bandwidth set $b_j = k\text{-NN}_X(m_j)$, then $\mu$ will $(\sqrt{n_{\max} \nu^2 + \eta A})$-interpolate $(X, y)$, where $n_{\max} = \max_{m_j \in M} |X_j|$.*

*Proof.* The $(\lambda, \ell, k)$-balanced assumption implies that at any point $x \in X$, then $k$ nearest neighbors in $M = \{m_1, \ldots, m_k\}$ (which are used to create the model $\mu$ at $x$) have $x$ in the set $X_j$ defined by each $m_j$. By the locally $(\nu, A, k)$-fit assumption, we know each local model $\mu_j$ will $(\sqrt{n_j \nu^2 + \eta A})$-interpolate on the data in $X_j$ including $x$, via Lemma 4.4.

Finally, we recognize that $\mu$ is a weighted average of the models associated with $M$ (that is $\mu = \sum_{j=1}^{k} p_j \mu_j$ for $\sum_{j=1}^{k} p_j = 1$). Hence, if each is at most $\sqrt{n_j \nu^2 + \eta A} \leq \sqrt{n_{\max} \nu^2 + \eta A}$ far from interpolating $x$, then the total interpolation error is at most $\sum_{j=1}^{k} p_j \sqrt{n_{\max} \nu^2 + \eta A} = \sqrt{n_{\max} \nu^2 + \eta A}$. $\qquad\square$

An important implication is that local KRR guarantees to be closer to interpolation than global KRR.

### 4.3 $\ell_2$-Convergence Rate of Local KRR

In this subsection, we show that local KRR– in addition to being locally adaptive, near-interpolating, and continuous – also can achieve standard learning bounds. Specifically we provide generalization error bound in an $\ell_2$ sense under some standard and mild assumptions. We provide two bounds depending on the algorithms behavior as the number of data points goes to infinity; we either consider (a) the number of local models is fixed and the number of data points per model goes to infinity, or (b) the number of local models also goes to infinity but the number of data points per model is fixed. In either case, we need to establish a few standard assumptions on the models built.

(A1): We assume we use clipped KRR for each model (Steinwart et al., 2009). That is, for a local data set $(X_i, y^{(i)})$ assume that $T_1 = \min\{y \in y^{(i)}\}$, $T_2 = \max\{y \in y^{(i)}\}$ that $T = T_2 - T_1$ and that if we predict the learned local model $\mu_i(x)$ returns a value outside of some $[T_1', T_2']$ where $T_2' \geq T_2$, $T_1' \leq T_1$, and $T_2' - T_1' < 2T$, then we clip to the nearest value in the range $[T_1', T_2']$; namely, $T_1'$ or $T_2'$. Note that this cannot increase the empirical error, and will not affect the near-interpolation property or the continuity property of our model.

(A2): Let $B_i = \{x \in \mathbb{R}^d \mid \|x - m_i\| \leq \lambda b_i\}$ be the ball on which the $i$th local model is learned, and for which its convergence properties will be well-defined. Local KRR requires $k$ local models (recall $k = \Theta(d)$ is a constant), to evaluate a query. Let $\mathcal{B}_k = \{x \in \mathbb{R}^d \mid |\{B_i \mid x \cap B_i \neq 0\}| \geq k\}$ be domain within at least $k$ such balls, and is where the local models are defined. We assume our data is drawn from a measure $\sigma$ whose support is contained in $\mathcal{B}_k$. Note that the way we determine $M$ (ensuring $X \subset \mathcal{B}_t$ with $t \geq k$) in Section 3 guarantees that the support of $\sigma$ is in $\mathcal{B}_k$ as the number of models $|M|$ goes to infinite with $n$. And in the case where $M$ is fixed we assume this has been achieved.

(A3): Assume $f^* : \mathbb{R}^d \to \mathbb{R}$ is a true generating model of our data $(X, y)$ and that $f^* \in L_2(\mathbb{R}^d) \cap L_\infty(\mathbb{R}^d)$, where $L_p$ are standard Lebesgue spaces in $\mathbb{R}^d$. As a smoothness assumption we make one of two refinements. Either (A3.1) we assume $f^* \in B_{2,\infty}^s(\sigma))$ where $B_{2,\infty}^s(\sigma)$ is a Besov-like space with smoothness parameter $s$; see Eberts & Steinwart (2013) for standard definitions of function spaces. Or (A3.2) we assume $y_i = f^* + \epsilon_i$, and $f^*$ satisfies a $(C_1, \alpha)$-Hölder condition s.t., $|f^*(x) - f^*(x)| \leq C_1 \cdot \|x - x'\|^\alpha$. We also assume the conditional variance of $\epsilon_i$ as $\mathsf{Var}(y_i \mid X = x_i)$ satisfies a $(C_1', \alpha')$-Hölder condition.

Next we observe that our global prediction $\mu$ is the weighted average $\mu(x) = \sum_{j=1}^{k} p_j \mu_j(x)$ of $k$ local predictions, where each $p_j \in [0, 1]$ and $\sum_{j=1}^{k} p_j = 1$. Hence if for true function $f^*$, each $\mu_i$ satisfies $\mathbb{E}\|\mu_i - f^*\|_2^2 \leq R_n$ for some diminishing error rate $R_n$ over the domain $B_i$, then we have

$$\mathbb{E}\|\mu - f^*\|_2^2 = \mathbb{E}\|\sum_j p_j \mu_j - f^*\|_2^2 = \mathbb{E}[(\sum_j p_j \|\mu_j - f^*\|_2)^2] \leq \mathbb{E}[(\sum_j p_j^2)(\sum_j \|\mu_j - f^*\|_2^2)] \leq k \mathbb{E}\|\mu_j - f^*\|_2^2 = k R_n,$$

so the global model $\mu$ also achieves that error rate, up to a constant factor $k$. Thus what remains is to show each local model achieves a certain error rate; it will then generalize to the global model. For the fixed

number of models, we use Eberts & Steinwart (2013)'s optimal KRR analysis, and as the number of models going to infinity we adapt Belkin et al. (2018a)'s analysis of a linear interpolation scheme.

**Fixed number of local models.** When we assume the number of local models $|M|$ is fixed, each local model observes at least $n'$ points, as $n$ goes to infinity, and hence $n'$ goes to infinity. Under assumptions (A1), (A2), and (A3.1) we can apply our settings to Theorem 3.1 and Corollary 3.2 in Eberts & Steinwart (2013), which gives the near-optimal (up to $\xi$) $\ell_2$ error rate for KRR.

**Lemma 4.6.** *For constant $C > 0$ and any $\xi > 0$, a local model $\mu_i$ observing $n'$ points $(X_i, y^{(i)}) \in \mathbb{R}^d \times \mathbb{R}$ from $\sigma, f^*$ satisfying (A3.1), and learning according to (A1), would have*

$$\mathbb{E}_{\sigma|B_i} \|\mu_i - f^*\|_2^2 \le C \cdot (n')^{-\frac{2s}{2s+d}+\xi}.$$

Applying the observation that local KRR is a weighted average of local models, and so inherit their convergence bound (up to constant $k$), we can state.

**Corollary 4.7.** *Consider for data $(X, y) \in \mathbb{R}^d \times \mathbb{R}$ from $\sigma, f^*$, under assumptions (A1), (A2), and (A3.1), as the number of data points $n$ goes to infinity, with a fixed set of local models each observing at least $n'$ data points. Then the model $\mu$ learned by local KRR satisfies, for some constant $C > 0$, and any $\xi > 0$,*

$$\mathbb{E}_{\sigma} \|\mu - f^*\|_2^2 \le kC \cdot (n')^{-\frac{2s}{2s+d}+\xi}.$$

**Number of local models goes to infinity.** When the number of points in each local model is fixed, but $n$ goes to infinity, so the number of local models goes to infinity as well, then we adapt the approach of Belkin et al. (2018a) that considered a linear interpolation scheme. They assume the strong Hölder smoothness in (A3.2), and consider what happens as $\delta_i$, the radius of each local piece, goes to 0. Their approach shows that with (A3.2) as $\delta_i$ is small enough, then a constant prediction for each region is sufficient, and we can use that $T$ is bounded in (A1) to show each local model $\mu_i$ is close to the constant function. Their Theorem 3.2 bounds $E\|\mu - f^*\|$ using 4 terms. Similar to how they observe in Corollary 3.3 as $n \to \infty$ three of those terms go to 0 in our setting: the first since by (A2) we assume the support of $\sigma$ is in $\mathcal{B}_k$, and the second and third since by smoothness (A3.2) and bounded $T$ in (A1) that as $\delta_i = \lambda b_i \to 0$, then our prediction $\mu_i$ is always within $T$ of the constant function, which is good enough. Taking the weighted average of $k$ local models, only the variance $(y_i - f^*(x_i))$ remains, as follows:

**Lemma 4.8.** *Consider for data $(X, y) \in \mathbb{R}^d \times \mathbb{R}$ from $\sigma, f^*$, under assumptions (A1), (A2), and (A3.2). For $\delta_{\max} = \lambda \max_i b_i$, the model $\mu$ learned by local KRR satisfies*

$$\mathbb{E}_{\sigma} \|\mu - f^*\|_2^2 \le \frac{2k}{2+d} \mathbb{E}_{x_i \sim \sigma}[(y_i - f^*(x_i))^2] + kC_1^2 \mathbb{E}[\delta_{\max}^{2\alpha}] + \frac{2k}{d+2} C_1' \mathbb{E}[\delta_{\max}^{\alpha'}].$$

*As the number of data points $n$ and number of local models goes to infinity, $\delta_{\max} = \lambda \max_i b_i \to 0$, we have*

$$\mathbb{E}_{\sigma} \|\mu - f^*\|_2^2 \le \frac{2k}{2+d} \mathbb{E}_{x_i \sim \sigma}[(y_i - f^*(x_i))^2].$$

# 5 Experimental Results

We examine the performance of local KRR against alternative models, and evaluate by several metrics. For the examples for $d = 2$ we have ground truth values on a fine grid $(X_G, y_G)$, in addition to input data $(X, y)$.

**Prediction Errors:** For tasks in $\mathbb{R}^2$ it is useful to plot the error $e_i = \mu(x_i) - y_i$ over $(x_i, y_i) \in (X_G, y_G)$ on a fine grid, with blue as a positive error, red as a negative error, and white near 0 error, in Figure 2.

**RMSE:** The root mean square error, $\text{RMSE} = \sqrt{\frac{1}{n} \sum_{x_i \in X_G} (\mu(x_i) - y_i)^2}$. This is measured over a fine grid $(X_G, y_G)$ or test data.

**Worst Case Error:** The worst case error, denoted $\ell_\infty$, shows how far the model is from interpolating the true data $(X_G, y_G)$: $\ell_\infty = \max_{x_i \in X_G} |\mu(x_i) - y_i|$.

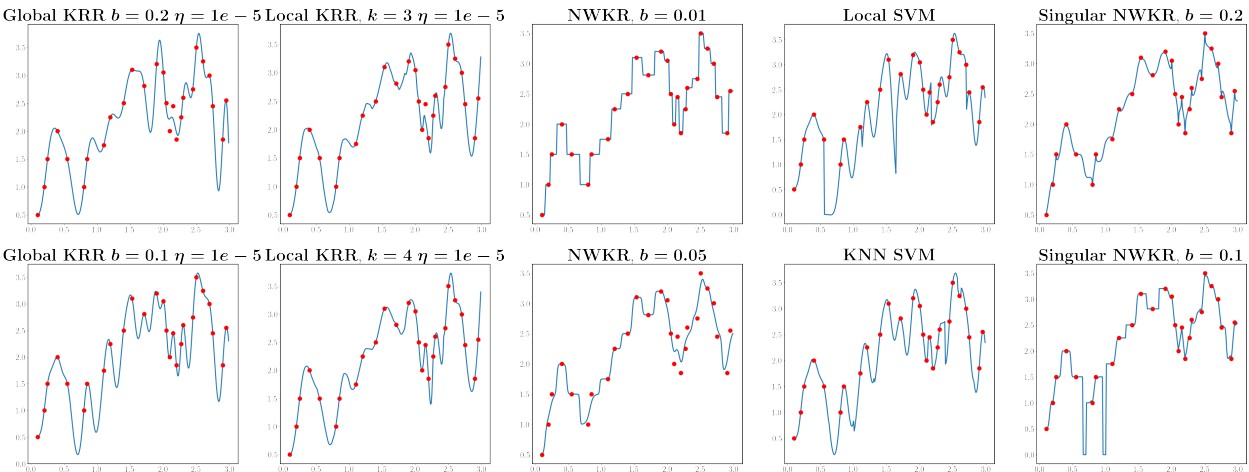

Figure 1: Fit models on synthetic data in $d = 1$.

**Relative Error:** Relative error $\varepsilon_i = \frac{\|y_i - \mu(x_i)\|}{\|y_i\|}$ is relevant for our $d = 3$ physics simulation example. We show maximum over a held out set.

**Average Curvature:** This captures the smoothness of the model for $d = 1$. The discrete curvature at a point $(x_i, y_i)$ can be defined $C_i = \frac{\|x_i'' y_i' - x_i' y_i''\|}{((x_i')^2 + (y_i')^2)^{3/2}}$, where $x_i'$, $y_i'$ are discrete derivatives, and $x_i''$, $y_i''$ are the discrete second derivatives. The Average Curvature avgC is the average on all grid points.

**Tuning Hyper-parameters.** For clarity we define several variables for local KRR, but some (like $\lambda$ and $k$) do not noticeably affect the result, other than in runtime, after their constraints are met. There are two modeling parameters that one could tune: the points per local region $\ell$ to control how large the local models should be, and ridge parameter $\eta$ to control how close to interpolating the data. Except in illustrative 1d examples, we choose these via grid search on a training set using RMSE. We consider $\ell \in \{10, 20, 30, 40, 50\}$ and $\eta \in \{1\text{e-7}, 1\text{e-5}, 1\text{e-3}, 1, 10\}$. For global KRR (build one single global regression model using KRR), singular kernel method, and NWKR (with Gaussian) we select bandwidth parameter $b$ from $b \in \{0.01, 0.05, 0.1, 1, 5\}$, and global KRR also choose $\eta$ from $\{1\text{e-7}, 1\text{e-5}, 1\text{e-3}, 1, 10\}$. In the real data experiment of methane data, we also compare knn-svm (Hable, 2013) and local-svm (Meister & Steinwart, 2016) with our methods. For knn-svm, we tune $k \in \{10, 20, 30, 40, 50\}$ and $\eta, b$ in the same ranges as local KRR. For local-svm we tune, *on each local model*, radius $r \in \{0.05, 0.1, 0.2, 0.3, 0.4\}$ and $\eta$ and $b$ as elsewhere. For RBF-Interpolator compared in 2D simulation example, we tune number of neighbors in $\{10, 20, 30, 40, 50\}$. We observe in all case that the selected values are typically near the median one, and do not largely affect the evaluation in the middle of these ranges.

### 5.1 1D Simulation

As an illustrative warm-up, we compare for $d = 1$ our method (local KRR) with singular kernel NWKR (Belkin et al., 2019b), NWKR with Gaussian kernel, global KRR with Gaussian kernel, Local-SVM (Meister & Steinwart, 2016) and KNN-SVM (Hable, 2013). The first five points match those in (Belkin et al., 2019b), and we extend more to show varying density. In Figure 1, the $\eta$ is the ridge parameter; the $b$ is the bandwidth; the $a$ is the exponent parameter in singular kernel (Belkin et al., 2019b). We calculate the avgC for all the fitted curves shown in the Figure 1 and the average curvature values from top-left to bottom-right are $\{12.93, 11.85\}$ for global KRR. $\{12.05, 11.01\}$ for local KRR, $\{171.99, 12.19\}$ for Gaussian-NWKR, $\{17.79, 18.18\}$ for local SVM and KNN SVM, and $\{20.07, 133.14\}$ for singular-NWKR, Our local KRR interpolates the data, and is the most smooth one with smallest avgC $= 11.01$.

The singular-NWKR interpolates the data, but exhibits strange artifacts of either cusps at the points if bandwidth too small, or blocky gaps if bandwidth too large. We attempted to tune parameters to remove these, but could not; these artifacts are even more prominent in the examples in their paper (Belkin et al.,

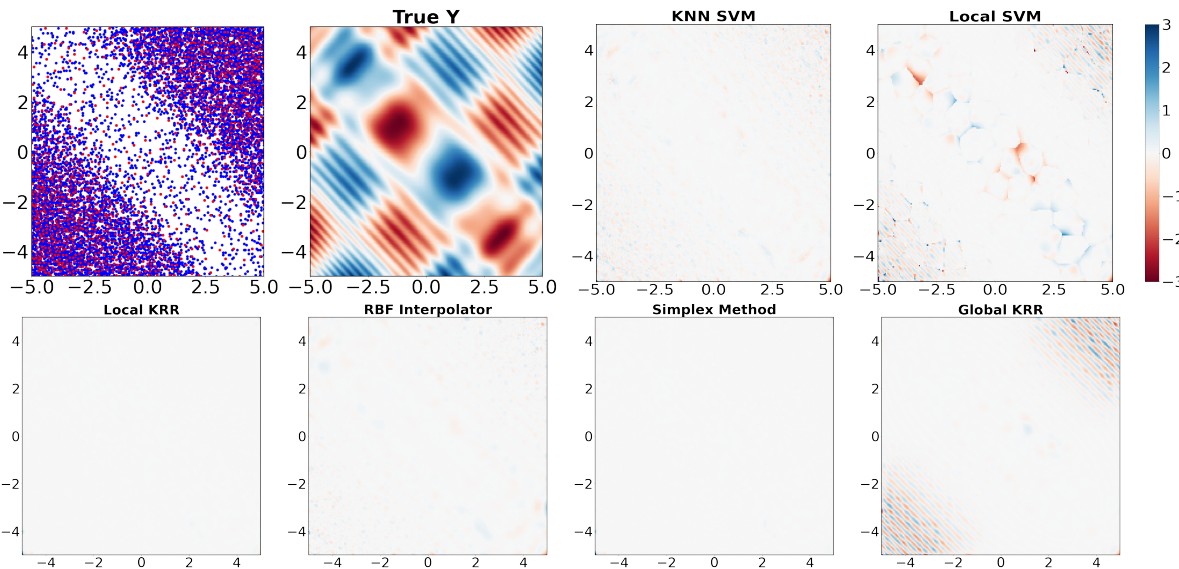

Figure 2: Synthetic data. First line, L→R: input data, true model, then prediction errors on knn-svm, local-svm. Second line, L→R: prediction errors on local KRR, RBF-Interpolator, simplex, global KRR.

2019b). The second case also has large average curvature. The Gaussian-NWKR either does not interpolate the data (bottom figure) or interpolates but is similarly blocky and has high curvature (top figure). The global KRR may not interpolate the data (top figure), but in this small example can be tuned to perform reasonably well (bottom figure). As the number of data points $n$ grows, it will be harder to interpolate; see Lemma 4.4. We used the following parameters: $k = 3$, $\eta = $ 1e-5, $b = 0.1$ for knn-svm and for local-svm radius $r = 0.5$, $\eta = $ 1e-5, $b = 0.1$. One can observe strange behavior in between the data points: large drops or discontinuities.

## 5.2 2D Simulation

As we discussed in Section 3, local KRR can adapt to the local density of the data. We design a 2D experiment where $x_1$ and $x_2$ are explanatory variables, and a response variable is generated with an undulating pattern, shown in Figure 2. The input data changes in density as the response variable requires more resolution More specifically, we place a fine grid with a data point $(x_1, x_2) \in [-5, 5] \times [-5, 5]$ every 0.05, which is 201 by 201 grid. The training data is selected using i.i.d. Bernoulli trials with probability equal to

$$p(x_1, x_2) = \max\{0.05, \log_{10}[1 + \log_{10}(1 + \frac{1}{2} \cdot (\frac{x_1}{2} + \frac{x_2}{2})^4)]\}.$$

And the response variable $y$ is defined as:

$$z_1 = \frac{\sqrt{2}}{2}x_1 + \frac{\sqrt{2}}{2}x_2, \;\; z_2 = -\frac{\sqrt{2}}{2}x_1 + \frac{\sqrt{2}}{2}x_2,$$

$$y = \sin(2z_1|z_1|) + \sin(\frac{1}{2}z_2|z_2|) + 5\cos(z_1)\sin(z_2).$$

In Figure 2(left), the blue points are training data $(x_1, x_2)$'s chosen by Bernoulli trials, and the red points are model points. Model points are denser when the training data is dense (i.e. in the upper right and bottom left corner). The second plot shows $y$ changes much faster in the top-right and bottom-right regions (the blue color means positive values of $y$, the red color means negative values). The data density distribution is similar to the variation of the function value in $y$.

We compare methods on this 2D synthetic dataset, showing the prediction errors on grids (Figure 2), and RMSE and $\ell_\infty$ error in Table 1. This experiment also shows Belkin et al. (2019b)'s Simplex Method and python's

|  | knn-svm | local-svm | local KRR | global KRR | RBF-Interpolator | simplex method |
|---|---|---|---|---|---|---|
| RMSE | 0.098 | 0.225 | 0.021 | 0.248 | 0.063 | 0.024 |
| $\ell_\infty$ | 3.17 | 9.67 | 2.24 | 3.76 | 2.44 | 2.73 |

Table 1: RMSE and $\ell_\infty$ error on 2D synthetic data.

RBF-Interpolator which does not have continuity guarantees. To tune hyper-parameters, we randomly select 10 percent of the data from training data as validation, then perform grid-search over hyper-parameters space. We use the best performed hyper-parameters on that validation set to evaluate the whole 201 by 201 grid. Details on a more thorough sensitivity analysis for local KRR is in Appendix B. Local KRR performs better than global KRR, and much better when the input data is sparse (in the middle). While global KRR and local SVM demonstrates significant error in many regions, for the interpolation-based ones (local KRR, RBF-Interpolator, and simplex) the error is mainly concentrated near the boundaries. We also note the simplex method, even in 2d where the complex size can be linear, it is about 40 times slower than local KRR.

### 5.3   Real Data: Slovakian Precipitation

We next compare local KRR with global KRR on a benchmark GIS dataset of Slovakia precipitation (Neteler & Mitasova, 2008). We choose up to 16000 training points randomly – beyond that size takes too long for global KRR, and is already too expensive (more than 12 hours) for the Simplex Method – retaining 180,000 for testing. Figure 3(left) shows visually that local KRR has less extreme prediction errors than global KRR. We also plot the RMSE and $\ell_\infty$ error as a function of training size in Figure 3(right), averaged over 5 trials, showing 1 std range. Local KRR has the least error under RMSE for all sample sizes, and always near the least error and the least at large sample sizes compared to any other method under $\ell_\infty$. Although local-svm and knn-svm are conceptually similar, they perform worse empirically.

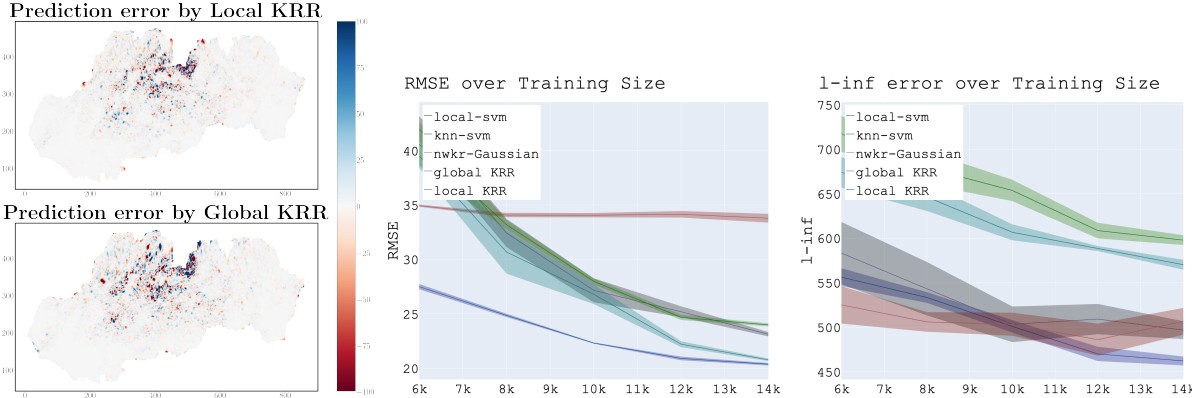

Figure 3: L→R: Prediction error, RMSE, and $\ell_\infty$ error for Slovakian Percipitation

### 5.4   3D Combustion Simulation

We next apply these techniques to a physics-driven combustion simulation. The explanatory variables $X$ measure the mass fraction and temperature, and the response variables $y$ capture the so-called source terms required to perform physics simulations. The data was gathered from a high-fidelity simulation of methane using the gri12 mechanism (Frenklach et al., 2021) in Spitfire (Hansen et al., 2020), and then its dimension reduced to 3 using a linear transform (Sutherland & Parente, 2009). This sort of example directly motivates the model we study: the data has variable density and requires locally different resolution, the data is of high accuracy so the model should basically interpolate the data, and such a model is to be used in simulating a PDE so is required to be smooth. Figure 4 shows the simulation lines and variable density. As such we use relative error to show how the model fits the observed data in a way that adapts to the local scale. We still build a single set of model parameters $\alpha$ for each local model.

The methane data set has $n = 44,000$ observations. We evaluate under a 67/33 train-test split under five different random seeds then, report the average and the standard deviation of the maximum relative errors (max rel. er) from five trials. We compare our local KRR with 4 other methods including global KRR, NWKR, Simplex method, singular kernel method, knn-svm and local-svm. Although all methods can achieve high r-squared value greater than 0.95, in this combustion task, the maximum relative error is the more important metric to guarantee that PDE simulations will work. From the results in the Table 2, we observe local KRR data has 1 order of magnitude less maximum relative error than the second best method, simplex method (which takes about 10x as long as local KRR in this setting). Compared to other methods, the improvement in error is 2 to 8 orders of magnitude. This shows how local KRR model is efficient, continuous, can adapt to data locally, and nearly interpolate.

|  | max rel.er | (std dev) |
|---|---|---|
| local KRR | **5.58** | (1.10) |
| global KRR | 2.01 e6 | (4.70 e5) |
| NWKR (Gaussian) | 7.26 e8 | (1.15 e8) |
| NWKR (Singular) | 2784 | (52.2) |
| Simplex | 66.35 | (37.81) |
| knn-svm | 4845 | (828) |
| local-svm | 719 | (69.0) |

Table 2: Maximum relative error for combustion data, averaged over 5 trials, and standard deviation.

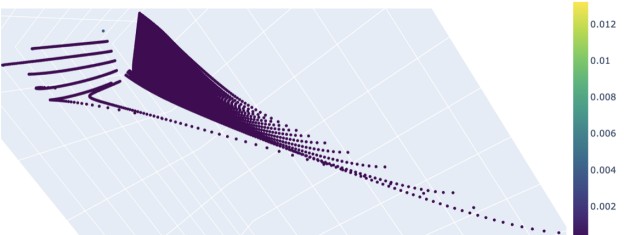

Figure 4: Data points for combustion data, with varying density, color-coded by relative error (mostly near 0) from local KRR.

## 6 Discussion

**Main novelties.** The main novelty of our paper is that it efficiently learns a locally-adapted regression function that is (a) continuous and (b) nearly interpolates the training data. Most ML work does not strive for either of these goals, but they are natural, especially in modern over-parameterized settings. For instance, similar divide-and-conquer kernel methods are not careful to blend local models together and either have discontinuities at model boundaries, or do not nearly-interpolate points near these boundaries. Other methods that meet these goals (like singular NWKR or high-degree polynomial regression) do not adapt to the local data properties, or (like building a simplicial complex) are very expensive.

**Nearly interpolation property.** The nearly interpolation goal is to have a local model precisely fit the local data, with error tolerance for each interpolation points controlled by $\nu$. Indeed the dependence on $n_{\max}$ in our Theorem 4.5 shows that *this is significantly easier for a locally-learned model (where n is small or a constant) than a global model.* For not every data set can KRR achieve small interpolation error ($\nu$) without a large ridge parameter $\eta$, and the data parameter $A$ precisely characterizes this.

**Time complexity and high dimensions.** The time complexity is as follows where $T_N$ is the (practically efficient, even in high dimensions) time for nearest neighbor search on $N$ data points. Choosing model points takes $O(n(k + T_n))$ time with a priority queue. Building models takes $O(kn(T_n + \ell^2))$ time if on average a point is in $O(k)$ models. Making a query/prediction takes time $O(T_{|M|} + k\ell)$. As $k = O(d)$, that is linear in dimension, and we can make the number of points in a model $\ell = O(1)$, hence *these are very efficient, even in high dimensions*

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

# A Proofs for Continuity

We provide the proof that the interpolation proportions are Lipschitz continuous.

Recall, that $r_j$ is the distance from $q$ to the $j$th closest model point. Recall $\bar{R}_k = \frac{1}{k}\sum_{i=1}^{k}(r_k - r_i)$, and let $R_k = \sum_{i=1}^{k}(r_k - r_i)$. And recall the $j$th weight is $w_j = (r_k - r_j)/(r_k - r_1)$. And the sum of all weights is $W = \sum_{j=1}^{k} w_j$ The proportion $p_j$ for model $\mu_j$ in the weighted average is normalized by the total weight, and is written

$$p_j = \frac{w_j}{W} = \frac{r_k - r_j}{r_k - r_1}\frac{(r_k - r_1)}{\sum_{i=1}^{k}(r_k - r_i)} = \frac{r_k - r_j}{R_k}.$$

Now we need to consider moving the query point $q \in \mathbb{R}^d$ by some small amount $\delta \in \mathbb{R}$. To formalize this consider some other point $q' = q + \delta u$, where $u \in \mathbb{R}^d$ and $\|u\| = 1$. So the perturbation is a change in $\mathbb{R}^d$, but we will only care about the magnitude $\delta$.

**Lemma A.1** (Lemma 4.1). *Consider moving the query point $q$ a distance at most $\delta \leq \frac{\bar{R}_k}{4}$. The proportion of model $\mu$ comprised of model $\mu_j$, denoted $p_j$, changes by at most $4\delta/\bar{R}_k$.*

*Proof.* Under the $\delta$-perturbation, each $j$th distance $r_j$ is updated to a value $r'_j$ in the range $[r_j - \delta, r_j + \delta]$. So $(r_k - r_j)$ changes by at most $2\delta$, except for $(r_k - r_k)$ which is always 0. Note this is true of the $j$th distance even if the sorted order of the models change. It is also true of the term effecting the contribution of the model that used to be ranked $j$, even if its ranking changes during this smaller perturbation of $q$.

Given the model $\mu_j$ is a $p_j$ proportion of model $\mu$ before the change, let $p'_j$ be the proportion after the change. Then we can analyze

$$\begin{aligned}
p_j - p'_j &= \frac{r_k - r_j}{\sum_{i=1}^{k-1}(r_k - r_i)} - \frac{r_k - r_j \pm \delta}{\sum_{i=1}^{k-1}(r_k - r_i \pm 2\delta)} \\
&= \frac{r_k - r_j}{R_k} - \frac{r_k - r_j \pm 2\delta}{R_k \pm (k-1)2\delta}.
\end{aligned}$$

We first get an upper bound

$$\begin{aligned}
p_j - p'_j &\leq \frac{r_k - r_j}{R_k} - \frac{r_k - r_j - 2\delta}{R_k + (k-1)2\delta} \\
&= \frac{r_k - r_j}{R_k}\frac{R_k + (k-1)2\delta}{R_k + (k-1)2\delta} - \frac{r_k - r_j - 2\delta}{R_k + (k-1)2\delta}\frac{R_k}{R_k} \\
&= \frac{R_k(r_k - r_j) + (r_k - r_j)(k-1)2\delta - (r_k - r_j)R_k + 2\delta R_k}{R_k(R_k + (k-1)2\delta)} \\
&= \delta 2\frac{(r_k - r_j)(k-1) + R_k}{R_k(R_k + (k-1)2\delta)}
\end{aligned}$$

Now using that $(r_k - r_j) \leq \frac{1}{j}\sum_{i=1}^{k}(r_k - r_i) \leq \frac{1}{j}R_k$ we can show

$$
\begin{aligned}
p_j - p'_j &\leq \delta 2 \frac{(r_k - r_j)(k-1) + R_k}{R_k(R_k + (k-1)2\delta)} \\
&< \delta 2 \frac{(r_k - r_j)(k-1) + R_k}{R_k(R_k)} \\
&= 2\delta \frac{(r_k - r_j)(k-1)}{R_k^2} + 2\delta \frac{1}{R_k} \\
&\leq 2\delta \frac{\frac{k-1}{j}R_k}{R_k^2} + 2\delta \frac{1}{R_k} \\
&\leq 2\delta \frac{k-1}{jR_k} + 2\delta \frac{1}{R_k} \\
&= 2\delta \frac{(k-1)/j + 1}{R_k} \leq 2\delta \frac{k}{R_k} = 2\delta/\bar{R}_k.
\end{aligned}
$$

Similarly the lower bound, using that $\delta \leq \bar{R}_k/4$ implies $2k\delta \leq R_k/2$, it can be shown as

$$
\begin{aligned}
p'_j - p_j &\leq \frac{r_k - r_j + 2\delta}{R_k - 2(k-1)\delta} - \frac{r_k - r_j}{R_k} \\
&= \frac{r_k - r_j + 2\delta}{R_k - 2(k-1)\delta} \frac{R_k}{R_k} - \frac{r_k - r_j}{R_k} \frac{R_k - 2(k-1)\delta}{R_k - 2(k-1)\delta} \\
&= \frac{R_k(r_k - r_j) + 2\delta R_k - (r_k - r_j)R_k + 2(r_k - r_j)(k-1)\delta}{R_k(R_k - 2(k-1)\delta)} \\
&= 2\delta \frac{(r_k - r_j)(k-1) + R_k}{R_k(R_k - 2(k-1)\delta)} \\
&\leq 2\delta \frac{\frac{k-1}{j}R_k + R_k}{R_k(R_k - 2(k-1)\delta)} \\
&= 2\delta \frac{\frac{k-1}{j} + 1}{R_k - 2(k-1)\delta} \\
&\leq 2\delta \frac{k}{R_k - 2k\delta} \\
&\leq 2\delta \frac{k}{R_k - R_k/2} = 4\delta/\bar{R}_k. \qquad \square
\end{aligned}
$$

And then we need to provide the Lipschitz-continuity result applied to the full local KRR model.

**Theorem A.2** (Theorem 4.3). *Assume for any $\mu_j \in M$, for $q, q' \in \mathbb{R}^d$ with $\|q - q'\| = \delta \leq \mathbb{R}_k/4$ and $|\mu_j(q') - \mu_j(q)| \leq L \cdot \delta$ and $\mu_j(q) \in (-T, T)$, then $|\mu(q') - \mu(q)| \leq \frac{4\delta(k-1)T}{\bar{R}_k} + L\delta$.*

*Proof.* We start with upper bound,

$$
\begin{aligned}
\mu(q') - \mu(q) &= \sum_{i=1}^{k-1} (p'_j \mu'_j(q) - p_j \mu_j(q)) \\
&\leq \sum_{i=1}^{k-1} (p'_j (\mu_j(q) + L\delta) - p_j \mu_j(q)) \\
&= \sum_{i=1}^{k-1} (p'_j - p_j)\mu_j(q) + L\delta \\
&\leq \sum_{i=1}^{k-1} |(p'_j - p_j)||\mu_j(q)| + L\delta \\
&\leq \sum_{i=1}^{k-1} \frac{4\delta}{\bar{R}_k}|\mu_j(q)| + L\delta \\
&\leq \frac{4\delta(k-1)T}{\bar{R}_k} + L\delta.
\end{aligned}
$$

Then similarly for the lower bound we have:

$$
\begin{aligned}
\mu(q') - \mu(q) &\geq \sum_{i=1}^{k-1} (p'_j (\mu_j(q) - L\delta) - p_j \mu_j(q)) \\
&\geq \sum_{i=1}^{k-1} -|(p'_j - p_j)||\mu_j(q)| - L\delta \\
&\geq -\frac{4\delta(k-1)T}{\bar{R}_k} - L\delta.
\end{aligned}
$$
$\qquad\square$

## B   Hyper-parameter Sensitivity Analysis of 2D Simulations

We provide a sensitivity analysis of hyper-parameters of our local KRR method in Table 5. It is based on a 10-fold cross-validation on training data for the 2D simulation in Section 5.2. In that 10-fold cross-validation, each time we pick 10 percent of training data as the validation set, build the model based on the rest of the training data, and record the RMSE on the 10 percent predictions. We perform this procedure 10 times for random scenario and report the average RMSE.

We observe that there is very little variation with the choice of $k$ (default is $k = d + 2$ and continuity requires $k \geq d + 1$) and $\lambda$ (default is $\lambda = 3$, following literature). However, $\ell$ is somewhat sensitive. This is the main scale parameter that controls how many data points are required to measure a distinct patch of the model. If $\ell$ is too small, the modeling in that patch may be too noisy, and if $\ell$ is too large, it may miss numerous small features. This is a problem dependent parameter – balancing the noise vs. the size of features – and so may not be surprising it benefits from some tuning.

| $k$ | RMSE |
|---|---|
| 2 | 0.035 |
| 3 | 0.034 |
| 4 | 0.031 |
| 5 | 0.032 |

Table 3: RMSE with fixed $\lambda = 4$, $\ell = 20$, tuned $k$

| $\ell$ | RMSE |
|---|---|
| 10 | 0.055 |
| 20 | 0.031 |
| 30 | 0.056 |
| 40 | 0.087 |
| 50 | 0.134 |

Table 4: RMSE with fixed $\lambda = 4$, $k = 4$, tuned $\ell$

| $\lambda$ | RMSE |
|---|---|
| 2 | 0.031 |
| 3 | 0.031 |
| 4 | 0.032 |

Table 5: RMSE with fixed $k = 4$, $\ell = 20$, tuned $\lambda$

