# OpenReview forum: "Local Kernel Ridge Regression for Scalable, Interpolating, Continuous Regression"
_TMLR — Accepted by TMLR_

### Review · Reviewer_DPQ8 · 2022-08-03

**Summary Of Contributions:**

In recent machine learning studies, the interpolation regime has gained more attention after seeing the well-known double descent phenomenon.
The interpolation is also interesting from the perspective of modeling physical dynamics that may not admit a nice closed-form formula.
Despite the classical machine learning paradigm paying more attention to the generalization perspective, this paper seeks a better interpolator with a localized version of kernel ridge regression.
The proposed model integrates multiple local kernel ridge regressors that change the kernel bandwidth adaptively to the local data density.
This approach is effective compared with the global model, particularly when the number of total data grows.
The local kernel ridge regression also has nice properties such as continuity (with respect to a query point) and guarantee on the (pointwise) interpolation error.
Lastly, the authors provide extensive experiments to see that the local kernel ridge regression indeed has better curvature and interpolation error globally on synthetic and real datasets.

**Broader Impact Concerns:**

This section does not apply to the current submission.

**Requested Changes:**

## Major requests

- **Hyperparameter sensitivity**: As mentioned in Weaknesses 2, having experiments to test hyperparameter sensitivity would be nice.
- **On general position (1)**: Some points could be made clearer. First, you can state its definition in Section 3.2 (when you first mention general position) or earlier; the definition is firstly stated in Section 4.1 in the current version, making the validation of the statement in Section 3.2 slightly time-consuming.
- **On general position (2)**: Second, its definition stated in Section 4.1 could be corrected: "no $d+2$ points are all equidistant from any query point" -> "given $d+1$ points, no point are equidistant from any other $d$ points". If I understand correctly, the latter corresponds to one of the standard definition of general position (no $d+1$ points are contained in any $d$-dimensional hyperplane). Let's consider the case $d=2$, then three ($d+1$) points contained in a straight line are not in general position (let's say this is *[condition A]*). Indeed, we can choose three points contained in a straight line such that the "middle" point is equidistant from the other two points. If we only preclude the case where four ($d+2$) points are not equidistant from each other, [condition A] is not necessarily satisfied. Hence, your definition does not seem to recover the standard one. Please correct me if I misunderstand.
- **On general position (3)**: Lastly, validation of the fact that $t \ge k \ge d+2$ implies $M_q$ is unique in Section 3.2 requires a little bit of heavy work (at least for me). The inequality $t \ge k$ was okay, but especially $k \ge d+2$ was a bit heavy (partly because the definition could be slightly incorrect as mentioned above). This part could be explained a little bit more.
- **Some explanation in Section 4.1**: The first paragraph of "Continuity of KNN Interpolation" was ambiguous. I would like to understand more what you refer to by "model point" and "the set of $k$ nearest neighbors" (neighbors of which point?), "boundary effect", and "jumping effect". Explaining what they refer to must make this paragraph easier to follow.

## Minor requests

- `\citet` and `\citep` can be used more appropriately. When you do not use a citation as a noun, `\citep` is a more appropriate choice. For example, citations in the fourth paragraph of Section 1, the first paragraph of "Learning the local models" in Section 3.1, the second paragraph of "Continuity of KNN Interpolation" in Section 4.1, the first paragraph of Section 5.1, and throughout Section 5.2 to Section 5.4 are more natural to be written with `\citep`.
- The sixth line of "Kernel Ridge Regression estimator" in Section 2: "dual coefficient" -> "dual coefficients"
- In "Interpolation Learning" of Section 2, what the existing work has done is usually written in the past tense. For example, "Belkin et al. (2018b) establish a theoretical foundation" -> "established", "Follow on work (Belkin et al., 2019b) shows that" -> "Follow-up work (...) showed that", "Liang & Rakhlin (2020) explains how using" -> "explained" in the first paragraph.
- In the last line of the first paragraph of "Kernel Ridge Regression estimation" in Section 2, do you mean that there does not exist a prior work about generalization for KRR with *moderately large ridge parameter* (namely $\eta$ is moderately large than zero), instead of $\eta = 0$ or small?
- The sixth line of "Fast Nearest Neighbor Search" of Section 2: "$p\_{th}$" -> "$p$th", to make it consistent with other parts of the paper.
- The second line of "Determine $M$" in Section 3.1: $c\_1$ -> $m\_1$? Also in the same line, do you consider the $\ell\_2$ hypersphere when you cover $\ell$ points around the chosen point $m\_1$? (please make it explicit)
- In the third paragraph of "Continuity of KNN Interpolation": "The Lipschitz factor will be larger" -> "will be smaller"? I'm looking at the statement of Theorem 4.3.
- In the third paragraph of "Continuity of KNN Interpolation", do you need to mention "the standard Delaunay complex"? Since many readers do not have background knowledge of it, this explanation may be distracting. If you feel necessary, I would recommend you add more explanation.
- Right after the proof of Corollary 4.2: "Lets" -> "Let us"
- In Theorem 4.3: Please make it explicit that the theorem assumes $M$ is $(\gamma,k)$-distributed.
- In the second paragraph of "Local Interpolation" in Section 4.2: $b\_i = \ell\mathrm{NN}(x\_i)$ -> $b\_i = \\|x\_i - \ell\mathrm{NN}(x\_i)\\|$
- In the ninth line of the fourth paragraph of "Local Interpolation" in Section 4.2, what do you refer to by "this" in "We define this as a radius..."?
- You do not seem to have defined "global KRR", which becomes a little bit confusing in the latter part of the paper because there are several confusing terminologies such as "global KRR", "local KRR", and "full local KRR".
- In Section 4.3, after "Fixed number of local models", the notation of expectations should be corrected to $\\mathbb{E}$ occasionally. In Lemma 4.6, the real value set should be denoted by $\\mathbb{R}$.
- In Figure 1, I prefer to see the name "singular NWKR" in the captions of the right-most column.

**Strengths And Weaknesses:**

## Strengths

1. **Motivation**: The authors claimed the importance of studying interpolation from the perspective of recent advances in overparametrization and physical modeling. The introduction is well-written and motivating.
2. **Simplicity**: The proposed local model is simple in terms of practical implementation, which requires only choosing the center points of the local models fitted independently. In addition, simplicity leads us to understand the model behavior in a transparent way. Indeed, the analysis provided in Section 4 can be followed rather effortlessly.
3. **Continuity of the model**: Surprisingly, many localized kernel models do not exhibit a good continuity property and curvature. In contrast, the proposed model has nice Lipschitz continuity (with respect to a query point), which is practically important as well, particularly in physical modeling.

## Weaknesses

1. **Dependency on data dimension**: Although the authors stated that the focus of this paper is not extremely high-dimensional data, I cannot avoid mentioning that the proposed model requires computational time linearly proportional to the data dimension due to $k = \Theta(d)$ points chosen in the query phase. Perhaps, it is interesting to see the empirical behavior of the proposed model with $k \ll O(d)$, regardless of whether the chosen points $M$ are in general position or not.
2. **Many hyperparameters**: The proposed model increases the number of hyperparameters compared with the global kernel ridge regression. Specifically, the number of coverages $t$ (used when we choose the center points $M$), the number of nearest neighbors $\ell$ (used when we adaptively choose the kernel bandwidth), the expanding factor $\lambda$ (used when we form the base set on which a single local model is applied), and the number of nearest neighbors $k$ (used when we query a test point) are newly introduced. Whereas the authors claimed that most of the hyperparameters do not influence the empirical performance so much, I would recommend supporting this point by showing the insensitivity with some experimental results.

---

> ### Author Response · Authors · 2022-08-19
> **Reply to Major Requests**
>
> ## Sensitivity of hyperparameters:
>
> For the sensitivity of $\ell$, $\lambda$, and $k$, we used 2D synthetic data based on reviewer Reviewer j594's design (we used 200 by 200 grid for x by y in our paper, but in his code, he used 201 by 201 grid for x by y):
>
> Fixed k = 4, $\lambda = 4$
> |varied $\ell$|RMSE|
> |---|---|
> |10 | 0.072 |
> |15 | 0.041 |
> |20 | 0.052 |
> |25 | 0.061 |
>
> Fixed k = 4, $\ell = 15$
> |varied $\lambda$|RMSE|
> |---|---|
> |2 | 0.065 |
> |3 | 0.045 |
> |4 | 0.041 |
> |5 | 0.040 |
>
> Fixed $\lambda = 4$, $\ell = 15$
> |varied $k$|RMSE|
> |---|---|
> |2 | 0.042 |
> |3 | 0.040 |
> |4 | 0.041 |
> |5 | 0.043 |
>
> We can see that RMSE is not that sensitive to $\lambda$, $k$, or $\ell$.
>
>
>
>
>
>
> ## On general position (1 & 3)
> Thank you for refocusing our attention on this claim based on general position in Section 3.2.  That claim about $M_q$ being unique under our assumption was incorrect; however, luckily, it was also not needed.  If one selects the $k' \leq k$ uniquely defined nearest points and also any arbitrary subset of $k - k'$ equally close points (where no other points are closer), then the claims about the algorithms hold.  All of those $k' - k$ points will get a weight of $0$, so it does not matter which are selected.
>
> We simply remove this uniqueness claim in Section 3.2; it now reads:
> > Let $t \geq k \geq d+2$, as we will see in Section 4.1, this will allow us to provide results on continuity of the local krr model.
>
> The arguments about continuity, which are made with more care in Section 4.1, are unaffected.
>
> ## On general position (2)
> In computational geometry, *general position* is more of a concept than a specific definition.  There are many ways it could be defined -- such as no 3 colinear points in 2d, no 4 coplanar points in 3d, no 3 lines in 2d intersect at a common point, or no 4 points that lie on a circle in 2d.  Different geometric constructions require different, but similar, measure-zero properties to avoid strange behavior. The standard practice is to define the version required for the setting at hand.  In the case of Delaunay triangulations in d dimensions, the required notion is no d+2 points on a common sphere.  Note that this can be viewed to generalize no d+2 points on a hyperplane in d dimensions, since, fixing a tangent point then as the center (and hence also radius) of a sphere goes to infinity, it converges to hyperplane.  In our setting, this is also the version of general position we need. Ultimately, we feel the use and definition here is standard and correct for our setting.
>
>
> ## Some explanation in Section 4.1
> We removed the informal terms and began by pointing out that we consider continuously changing the location of the query $q$.  The next text reads:
>
> > Consider continuously moving the position of a query point $q$.  As any model point moves into the set of $k$ nearest to $q$, its weight is initially zero.  So there is no boundary effect discontinuity in the contribution of the nearest $k$ models when suddenly a different set of points are the $k$ nearest.  Also, when two points change their relative position in the sorted order, they have the same weight.  So again there is no discontinuity in the weight as the relative order of model points changes within that nearest $k$.

---

> ### Author Response · Authors · 2022-08-19
> **Reply to Minor Requests**
>
> ## Minor requests:
>
> - [x] citations changed from \citet to \citep
> - [x] the dual coefficient -> dual coeffcients
> - [x] make changes to past tense.
> - [x] We mean that (Liang & Rakhlin 2020) only considered for high dimension (d > n) setting and ridge term goes to 0. We were unaware of comparable work in low dimension (n > d) setting.
> - [x] change $p_{th}$ to $p$th in the sixth line of "Fast Nearest Neighbor Search" of Section 2
> - [x] change $c_1$ to $m_1$ in the second line of "Determine M" in sec 3.1
> - [x] In the third paragraph of "Continuity of KNN Interpolation": "The Lipschitz factor will be larger" -> "will be smaller"?  -> It should be "smaller" I think. Sorry for the typo.
> - [x] We added a citation to a well-known book devoted to Delaunay complex for the curious reader.  Since the remark about this is mostly an aside -- noting the connection for the interested reader -- we felt it would be distracting to elaborate too much.
> - [x] Right after the proof of Corollary 4.2: "Lets" -> "Let us" -> text updated
> - [x] In Theorem 4.3: Please make it explicit that the theorem assumes M is (\gamma, k) distributed. -> We updated it in Theorem 4.3.
> - [x] In the second paragraph of "Local Interpolation" in Section 4.2: change b_i = $\ell$NN(x_i) -> b_i = \|x_i - $\ell$NN(x_i) \| -> text updated
> - [x] In "Local Interpolation" the "this" meant the use of $\lambda b(k)$; -> text updated.
> - [x] You do not seem to have defined "global KRR", which becomes a little bit confusing in the latter part of the paper because there are several confusing terminologies such as "global KRR", "local KRR", and "full local KRR". -> We added the explanation in the sec 5.1, tunning hyper-parameter paragraph, on the sixth line.
> - [x] In Section 4.3, after "Fixed number of local models", the notation of expectations should be corrected to \ensuremath{\mathbb{E}} occasionally. In Lemma 4.6, the real value set should be denoted by \ensuremath{\mathbb{R}}. -> text updated
> - [x] Singular NWKR In Figure 1 ->  subtitles updated

---

### Review · Reviewer_j594 · 2022-08-05

**Summary Of Contributions:**

### Research Target
This paper considers a supervised regression with a special focus on the data interpolation.

### Overview of the method
This paper proposes a method called Local KRR, as the smoothed weighted average of multiple KRRs.
Let $D = \\{x_n, y_n\\}\_{n=1}^N$ be a training set.
Local KRR proceeds as follows.
First, we select a subset of input points $M \\subset \\{x_n\\}\_{n=1}^N$ so that all the inputs reside in the close neighborhoods of $M$.
Second, for each $m_j \in M$, we fit KRR $\mu_j$ using the data points in $D$ within the close neighborhood of $M$ so that $\\|m_j - x_n\\| \le \tau_m$ for some $\tau_m > 0$ (set as $3 \times$ the distance to the $\ell$-th nearest neighbor in the paper).
Third, at the prediction phase for the input $x$, the predictions made by each KRR $\mu_j(x)$ is averaged using a distance-based weight as follows.
Let $m_i \in M$ be the $i$-th nearest neighbor of the input $x$, and denote its distance to $x$ by $r_i = \\|m_i - x\\|$.
The weight for KRR $\mu_i$, corresponding to $m_i$, is chosen as $w_i = \frac{r_k - r_i}{r_k - r_1}$, where $k$ is the number of neighboring KRRs used for prediction.
The resulting predictions is then given as $\frac{\sum_{i=1}^k w_i \mu_i(x)}{\sum_{i=1}^k w_i}$.

### Some properties of Local KRR
In the paper, the authors reported that Local KRR has the following four properties.
1. it adapts to the local density
2. it is efficient to evaluate near data X
3. it is continuous
4. it (nearly) interpolates data points

The first property follows from the fact that each KRR is trained using data points in a small neighborhood.
Such a local model can adapt to the smoothness of the underlying function within the small neighborhood.
Some local models get very smooth if the underlying function is almost constant in the small neighborhood, while some other local models may become sharper if the underlying function changes dynamically in the small neighborhood.
Note that, it would be difficult to fit a function with both smooth and sharp local behaviors using a single model such as KRR and Nadaraya-Watson Kernel Regression (NWKR) with a kernel function with a common bandwidth.

The second property follows from the fact that the prediction of Local KRR is made by the weighted average of only $k$ KRRs.
Because each KRR is trained using a small number of points, the prediction made by each KRR is computationally fast, in contrast to global KRR that uses all the training points for prediction.

The third and fourth properties follows from the fact that the weight $w_i$ is continuous / Lipschitz continuous in the neighborhood of any input $x$.
The authors provided rigorous proofs for these properties.

### Experimental results
The authors performed experiments on 1D, 2D simulation data, and 2D and 3D real world data (Slovakian Precipitation, Combustion Simulation).
The results show that Local KRR performs better than some standard interpolation methods such as NWKR, KRR, and some of the neighbor-based local models.

**Broader Impact Concerns:**

There is no ethical concern.

**Requested Changes:**

### Request 1.
As I pointed out in Weakness 1, the performance of Local KRR seems to be inferior to `RBFInterpolator` of `scipy.interpolate`.
This raises a question of the effectiveness of Local KRR.
The selection of the baseline methods as well as their hyperparameter choices may require reconsideratoin.

### Request 2.
As I pointed out in Weakness 2, Local KRR can be interpreted as the interpolation of multiple KRRs using the Nadaraya-Watson estimator.
Because there are rich literatures on the Nadaraya-Watson estimator, exhaustive discussions will be essential, in particular, on the relationship with the prior studies in the literatures, and on the novelty of Local KRR over the existing literatures.
The current paper completely misses this relationship.

**Strengths And Weaknesses:**

## Strong aspects
One strong aspect of the paper would be the theoretical part showing that Local KRR has some good properties.
The continuity of Local KRR will be of particular importance in practice because some neighbor-based local models, such as nearest neighbor methods, are not continuous.

Another strong aspect of the paper would be the adaptivity and computational scalability of Local KRR.
Different from global KRR and NWKR, Local KRR does not require accessing the entire training set for each individual KRR both in the training and prediction phases: each KRR handles only the points in small neighborhoods.
This enables Local KRR to interpolate functions with varying smoothness.
Standard global KRR and NWKR have a difficulty of handling such functions because the bandwidth of the kernel function determines the global smoothness of the fitting function.
The computational scalability is also of importance in practice.
NWKR requires computing distances between the input $x$ and all the training points which takes $O(N)$ time, while global KRR requires $O(N^3)$ time for computing the inverse of the Gram matrix.
In Local KRR, only the neighbor search takes time dependent on $N$ (but only $o(N)$ time), and the training and predictions of each KRR takes time dependent only on the size of the local neighborhood, which is typically far smaller than $N$.


## Weak aspects
This paper has two major weakness, as follows.

### Weakness 1.
In the 2D simulation experiment, the proposed Local KRR performs worse than `RBFInterpolator` in `scipy.interpolate`.
This raises a question on the practical utility of Local KRR.

[Detail]

I used the `neighbor=10` and the default values for the other options in `RBFInterpolator`.
With the option `neighbor=10`, `RBFInterpolator` fits KRR for each $x$ using the 10 neighbors of the input $x$, similar to the proposed Local KRR.
It attained RMSE around 0.09, far smaller than the reported 0.747 of Local KRR.
Of course, in theory, `RBFInterpolator` with `neighbor` option is not continuous at the boundary where the membership of the neighbor changes.
However, given the far smaller RMSE, we will not be able to ignore `RBFInterpolator` just because it is not continuous.
In practice, accurate prediction may be more worthy than the continuity.

```
### This code produces RMSE = 0.09020604653847476 ###

import numpy as np
from scipy import interpolate

def prob(x):
    p = np.log10(1 + np.log10(1 + 0.5 * (0.5 * x[:, 0] + 0.5 * x[:, 1])**4))
    return np.maximum(0.05, p)

def response(x):
    z1 = x[:, 0] / np.sqrt(2) + x[:, 1] / np.sqrt(2)
    z2 = - x[:, 0] / np.sqrt(2) + x[:, 1] / np.sqrt(2)
    return np.sin(2 * z1 * np.abs(z1)) + np.sin(0.5 * z2 * np.abs(z2)) + 5 * np.cos(z1) * np.sin(z2)

# data generation
np.random.seed(0)
t = np.linspace(-5, 5, 201)
X1, X2 = np.meshgrid(t, t)
x_mesh = np.stack([X1.flatten(), X2.flatten()], axis=1)
y_mesh = response(x_mesh)
p = prob(x_mesh)
q = np.random.rand(p.size)
x, y = x_mesh[p >= q], y_mesh[p >= q]

# training and prediction
rbf = interpolate.RBFInterpolator(x, y, neighbors=10)
z_mesh = rbf(x_mesh)

# RMSE
print(np.sqrt(np.mean((y_mesh - z_mesh)**2)))
```

### Weakness 2.
Local KRR is actually a interpolation of multiple KRRs using the Nadaraya-Watson estimator.
This fact is completely overlooked in the paper.

[Detail]

Recall that Local KRR makes a prediction for the input $x$ as
$$
y = \frac{\sum_{i=1}^k w_i \mu_i(x)}{\sum_{i=1}^k w_i} = \sum_{i=1}^k p_i \mu_i(x), \quad \mathrm{where} \; p_i = \frac{w_i}{\sum_{i=1}^k w_i}, \quad w_i = \frac{r_k - r_i}{r_k - r_1}, \quad r_i = i\mathrm{-th} \min_{m \in M}\\|x - m\\| .
$$
The choice of the weight $w_i$ is identical to the triangular kernel with the bandwidth equals to $r_k$.
The triangular kernel is given as
$$
K(\\|m - x\\|) = \max\\{0, 1 - \\|m - x\\|\\} .
$$
The scaled version with the bandwidth $h > 0$ is given as
$$
\frac{1}{h} K\left(\frac{\\|m - x\\|}{h}\right) = \frac{1}{h^2}\max\\{0, h - \\|m - x\\|\\} .
$$
Using this triangular kernel, the interpolation of multiple KRRs using the Nadaraya-Watson estimator can be then expressed as
$$
y = \frac{\sum_{m \in M}\max\\{0, h - \\|m - x\\| \\} \mu_m(x)}{\sum_{m \in M}\max\\{0, h - \\|m - x\\|\\}}.
$$
Setting $h = r_k$ yields Local KRR because $\max\\{0, r_k - \\|m - x\\|\\}$ is nonnegative only for $\\|m - x\\| \ge r_k$, i.e, the first $k$ nearest neighbor of $x$.

This finding opens up several questions, as follows.

First, what is the advantage of using the triangular kernel?
Other kernels, such as Epanechnikov kernel and Quartic kernel, can also induce similar estimators using only the $k$ nearest neighbors.
That is, the proposed weight $w_i$ is not a unique choice for the purpose considered in the paper.

Second, some of the theoretical findings can be explained by the Nadaraya-Watson estimator.
For example, the continuity of the fitted function (one of the major contributions) would be trivial from the continuity of the Nadaraya-Watson estimator.
The interpolation property as well as the convergence rate reported in the paper can be explained as well.
For example, [Ref1] provides an asymptotic convergence of the Nadaraya-Watson estimator for interpolating multiple functions.

[Ref1] Kernel regression with functional response, Electric Journal of Statistics, 2011.

### Minor Weakness.
I found that Theorem 4.3 can be improved by omitting the factor $L \delta$.
$$
| \mu(q') - \mu(q) | = \left| \sum_{i=1}^{k-1} (p'_i \mu'_i(q') - p_i \mu_i(q)) \right|
$$

$$
\le \left| \sum_{i=1}^{k-1} (p'_i T - p_i T) \right|
$$

$$
\le T \sum_{i=1}^{k-1} \left| p'_i - p_i \right|
$$

$$
\le \frac{4 \delta (k - 1)T}{\bar{R}_k} .
$$
For this bound to hold, only the Lipschitz continuity of $p$ and the boundedness of $\mu$ will be sufficient.

---

> ### Author Response · Authors · 2022-08-19
> **Reply to Weakness 1**
>
> ## Weakness 1: RBF Interpolator
> Thanks for pointing out the RBF interpolator!  We will incorporate this into the revision; in the mean time, here are some comments.
>
> When running it on the 1D data set, one can indeed see some small discontinuities.  Although they are perhaps more subtle than other discontinuous methods, we think the lack of guarantees is important since it does not prevent more serious issues.
>
> Empirically, we only ran one trial on this 2D dataset with our default hyper-parameters choices.  And due to the randomness of training points, different trials may give different results. Our data design is also slightly different from what you showed in your code. We used np.linspace(-5, 5, 200) rather than np.linspace(-5, 5, 201).  We re-ran the code under the same data generation strategy as you suggested, comparing the RBF interpolator with our method (slightly changed the number of the evaluated models from 3 to 4), and we got a comparable result (around 0.07) in the code below.  Under different randomness, we only use 3 evaluated models we got 0.135,  improving on the 0.323 we got last run.
>
>
> ```
> ### rmse = 0.07192164953072741 ###
> import numpy as np
> from scipy import interpolate
>
> import faiss
> import numpy as np
> from numba import njit, jit
> from sklearn.kernel_ridge import KernelRidge
> def local_any_dimension(model_point, X, test_x, y, k, C1, C2, alpha, k2, sigma, linear = True):
>     n,d = model_point.shape
>     index = faiss.IndexFlatL2(d)
>     index.add(X)
>     model_map = {}
>     max_i = 0
>     for j in range(n):
>         D,I = index.search((model_point[j,:]).reshape((1,d)),k)
>         b = np.max([np.linalg.norm(model_point[j,:] - x) for x in X[I[0],:]])
>         l, D_1, I_1 = index.range_search((model_point[j,:]).reshape((1,d)), C1 * b ** 2)
>         new_x = X[I_1, :]
>         knn_y_i = y[I_1]
>         clf = KernelRidge(alpha=alpha, kernel = 'rbf', gamma = 1 / (C2 * b ** 2)).fit(new_x, knn_y_i)
>         model_map[j] = clf
>     m,d1 = test_x.shape
>     index_1 = faiss.IndexFlatL2(d)
>     index_1.add(model_point)
>     pred_y = np.zeros((m,d1))
>     for q in tqdm(range(m)):
>         D,I = index_1.search((test_x[q,:]).reshape((1,d)),k2)
>         I_S = I[0]
>         distances = np.array([np.linalg.norm(test_x[q,:] - x) for x in model_point[I_S,:]])
>         radius_min = np.min(distances)
>         radius_max = np.max(distances)
>         if radius_min != radius_max:
>           raw_weight = np.array([(radius_max - x)/(radius_max - radius_min) for x in distances])
>         else:
>           raw_weight = np.ones(k2)
>         if linear:
>           normalized_weight = raw_weight / np.sum(raw_weight)
>         else:
>           normalized_weight = np.exp(raw_weight * sigma) / np.sum(np.exp(raw_weight * sigma))
>         w = [model_map[s].predict(test_x[q,:].reshape(1,-1))[0] for s in I_S]
>         pred_y[q] = np.sum([w[i] * normalized_weight[i] for i in range(len(I_S))], axis=0)
>     return pred_y
>
> @jit
> def gon_clustering_aprox(X, k, k2, c):
>     n,d = X.shape
>     f = np.random.randint(0, n-1)
>     s = np.zeros((k, d))
>     centers = []
>     centers.append(f)
>     s[0,:] = X[f,:]
>     phi  = [0 for _ in range(n)]
>     r = []
>     index = faiss.IndexFlatL2(d)
>     index.add(X)
>     for t in range(n):
>          D,I = index.search((X[t,:]).reshape((1,d)),k2)
>          r.append(D[0][-1])
>     for i in range(1, k):
>         M = 0
>         s[i, :] = X[0, :]
>         new_center = 0
>         for j in range(n):
>             if j not in centers and np.sqrt(max(np.linalg.norm(X[j,:] - s[phi[j],:])**2 - r[centers[phi[j]]],0)) > M:
>                 M = np.sqrt(np.linalg.norm(X[j,:] - s[phi[j],:])**2 - r[centers[phi[j]]]**2)
>                 s[i,:] = X[j,:]
>                 new_center = j
>         centers.append(new_center)
>         for j in range(n):
>             if np.sqrt(max(np.linalg.norm(X[j,:] - s[phi[j],:])**2 - r[centers[phi[j]]],0)) > np.sqrt(max(np.linalg.norm(X[j,:] - s[i,:])**2 - r[centers[i]],0)):
>                 phi[j] = i
>     return phi, s
>
>
> def prob(x):
>     p = np.log10(1 + np.log10(1 + 0.5 * (0.5 * x[:, 0] + 0.5 * x[:, 1])**4))
>     return np.maximum(0.05, p)
>
> def response(x):
>     z1 = x[:, 0] / np.sqrt(2) + x[:, 1] / np.sqrt(2)
>     z2 = - x[:, 0] / np.sqrt(2) + x[:, 1] / np.sqrt(2)
>     return np.sin(2 * z1 * np.abs(z1)) + np.sin(0.5 * z2 * np.abs(z2)) + 5 * np.cos(z1) * np.sin(z2)
>
> # data generation
> np.random.seed(0)
> t = np.linspace(-5, 5, 201)
> X1, X2 = np.meshgrid(t, t)
> x_mesh = np.stack([X1.flatten(), X2.flatten()], axis=1)
> y_mesh = response(x_mesh)
> p = prob(x_mesh)
> q = np.random.rand(p.size)
> x, y = x_mesh[p >= q], y_mesh[p >= q]
>
> membership, model_points_2 = gon_clustering_aprox(x.astype(np.float32), 1500, 10, 1)
> preds = local_any_dimension(model_points_2.astype(np.float32),
>                             x.astype(np.float32),
>                             x_mesh.astype(np.float32),
>                             y, 10, 4, 1, 1e-3, 4, 0.05, True)
>
> print(np.sqrt(np.mean((preds[:,0] - y_mesh)**2)))
> ```

---

> ### Author Response · Authors · 2022-08-19
> **Reply to Weakness 2 and Minor Requests**
>
> ## Weakness 2: Connection to NWKR
>
> Thank you for pointing out the interesting connection to Nadaraya-Watson Kernel Regression.
> However, we would like to point out that this way of formulating the problem, it uses a variable-width triangle kernel, so the width of the kernel changes depending on which query point is used.  Most of the analyses of NWKR we are aware of (including the referenced EJS11 paper) consider a fixed kernel -- so do not directly apply to our setting.
>
> That said, we believe there would be a way to leverage this interpretation, and associated analysis, to obtain at least some of our results.  However, it seems to formalize such results this way, one would still need to analyze carefully how the bandwidth parameter varies as the query point changes, and then how this affects the kernel weighting scheme.  Indeed, this is the bulk of our analysis in Section 4.1 and Appendix A.
>
> ## Minor Request
>
> There is an issue with the new proof of Theorem 4.3 sketched by the reviewer.  In the second step, the expression is maximized by mapping one of the $\mu_i(q)$ terms to $-T$ (instead of $T$).  After factoring out a $T$ this leaves a $p_i' + p_i$,  instead of a $p_i' - p_i$ for which we have a bound.
>
> We suspect something like this can be made to work with an extra factor of $2$, but that in most cases the bound we provide (using the Lipschitz factor $L$) will be better.  So we have elected to retain our proof and bound.

---

### Review · Reviewer_Xqyg · 2022-08-15

**Summary Of Contributions:**

This paper proposes local kernel ridge regression, where the key idea is to perform kernel ridge regression to each local region determined based on k-NNs.
Although the proposed algorithm itself is simple, the authors thoroughly study its theoretical properties in terms of continuity, interpolation, and convergence.
In addition, the proposal is empirically examined on synthetic and real-world datasets compared to relevant regression methods.


**Broader Impact Concerns:**

I do not have any concerns.

**Requested Changes:**

Please refer to the Novelty, Quality 1, 3, and Presentation 2 in the above comment.

**Strengths And Weaknesses:**

## Novelty

The novelty of this paper is not convincing in the current submission. The proposed algorithm can be viewed as a variant of a combination of kNN regression and kernel regression, and I think such an idea has already been widely used. For example, I can easily find the R package (https://cran.r-project.org/web/packages/KernelKnn/KernelKnn.pdf). However, there is no discussion about the difference between the proposal and such related approaches in this paper. Although I think the contribution of the theoretical part of this paper is original and worth publishing, discussion of such related work is necessary.

## Quality

1. The proposal uses three levels of local search, selection of M, l-NN, and RANGE, and all of them are based on different search strategies. However, it is not clearly demonstrated why such three steps are required in the proposal. For example, how about just finding k-NN for each data point and applying Algorithm 2? I imagine that they may be necessary to obtain some theoretical properties or empirical effectiveness, while it is hard to see it in the current submission. Please clearly explain the validity of the proposed strategy.
1. Theoretical analysis is strong: it is thorough and convincing.
1. Empirical analysis is not thorough. In particular, only two simple synthetic datasets and one real-world datasets are used, which is not convincing. Since the proposal is a simple approach, it is interesting to see how it works on various types of datasets. I strongly recommend adding experiments on a number of real-world datasets (from small to large, low-dim to high-dim).

## Presentation

1. This paper is overall clearly written and easy to follow.
1. Presentation of Section 4 can be improved. Although the authors carefully explain    all the necessary materials, it is a bit difficult to understand the motivation of each theoretical result. In particular, it is not clear which part is a straightforward consequence from the kNN property itself, and which part is non-trivial due to the localness of the proposal. For example, is the first half of Section 4.2 ("Small noise, good fit assumption" and "Analysis of the small noise, good fit setting.") a direct consequence from the standard kNN? If so, they can be removed.

---

> ### Author Response · Authors · 2022-08-19
> **Reply to Requests: kernelKNN and High-d dataset**
>
> ## kernelKNN
> Thank you for pointing out the KernelKNN R package.  We already compared to a very similar method, knn-svm.  knn-svm identifies the k nearest neighbors and then learns the best weights on that subset of data, based on kernel ridge regression.  The proposed approach (kernelKNN) applies the standard NWKR method to this selection of k-nearest neighbors.  It is well-known that KRR tends to outperform NRKR, and our method consistently outperforms knn-svm empirically.  Moreover, both of these methods do not guarantee continuity, part of our desiderata.
>
> ## high-d data set
> As you suggested, we tried a larger dimension dataset pol, which has 15000 observations and 26 explanatory variables from https://github.com/treforevans/uci_datasets. 90 percent of data is treated as training data, and 10 percent of data as test data. Ridge parameter is fixed as $\eta$ = 1e-2. The $d$ in singular kernel NWKR is set as 20 (around the number of data dimensions).  Choices of bandwidth for global KRR, local SVM, NWKR, KNN-SVM, or Singular Kernel NWKR are selected from [0.1, 0.5, 1, 5, 10]. Choice of $\ell$ for local KRR are from [10, 50, 100, 150], $\lambda$ are from [1.5, 2, 2.5, 3], and $k = 3$. The choice of $K$ nearest neighbor of KNN-SVM are from [10,50,100,150]. We select hyperparameters based on the training RMSE and report test RMSE. The result is shown as below:
>
>
>
> |   |RMSE|
> |---| ---|
> |Global KRR| 12.035|
> |KNN SVM| 10.275 |
> |Local SVM| 10.376|
> |Singular Kernel|9.788|
> |NWKR | 10.857|
> |Local KRR| 9.192|
>
>
>
>
>
>
> Even though our method mainly focuses on the dataset with variable local density and moderate dimensions, the performance on this real dataset is also comparable to other candidate methods.

---

> > ### Comment · Reviewer_Xqyg · 2022-08-23
> > **My concerns are not solved yet**
> >
> > Thank you for your response. I still have concerns w.r.t. the following points:
> >
> > 1. There is no answer to my concern stated at "Quality 1" in my original review. It would be great if you could address this point.
> > 1. The answer to my novelty concern is unfortunately not convincing. I am aware that knn-svn is already compared in the submission, but an important point is that more thorough and careful discussion is required to show the novelty of the proposal.
> > 1. I have an additional concern of the sensitivity of hyperparameters stated in the authors' answer to Reviewer DPQ8. First, the authors mention that the RMSE is not sensitive to $\lambda$, $k$, or $l$, but it seems to be actually sensitive to $\lambda$ as the RMSE varies w.r.t. changes in $\lambda$. So this point should be carefully addressed. Second, and more importantly, the RMSE is almost invariant w.r.t. $k$, which means that the $M_q$ selection step using kNN has little effect in the proposed algorithm and may not be fundamental in practice. For example, what will happen if we directly use the entire $M$ instead of $M_q$? This question is actually related to my comment 1.

---

> > > ### Author Response · Authors · 2022-08-23
> > > **more comments**
> > >
> > > Thank you for your continued engagement with our paper.
> > >
> > > ## 1. Re: Quality 1
> > > I guess we are not quite sure what you are asking in this question.  Do you mean apply k-NN on the full set of data points X, or (as we already do in Alg 2) the model points M?  The reason we use the model points is then we can pre-compute any regression model on these local regions, and then just have to evaluate that constant number of models at query time.  This aids in efficiency (compared to querying neighbors, and then learning models around each nearby x in X).  We do not explore this so much in this paper, but we could also enforce M satisfy general position properties (via perturbation) or other more advanced coverage properties.
> > >
> > > However, you are right, one could just query the k nearest data points from X, and then build models around each of them.  If we use the same weighting scheme we could also show similar continuity properties.
> > >
> > > ## 2.  Re: Novelty
> > > While we think there are many aspects of novelty in our algorithm and its new analysis, we think the most clear way to make it clear is as follows.  That is, this is the first paper to study regression models that (1) adapts to local density, (2) has a continuous response function, and (3) (nearly) interpolates the training data.  We show empirically why point (1) is important over global models, and provide theoretical analysis on points (2) and (3).  Algorithms like knn-svm and kernelKNN regression do not satisfy point (2).
> > > We reached this desiderata from some applied collaboration, and feel they are meaningful in applications, intuitive, and mathematically interesting.
> > >
> > > ## 3a. Re: sensitivity of lambda
> > > As we mention in the paper, our choice of lambda=3 is a standard practice in many kernel methods.  This has been explored extensively in the literature on for instance kernel density estimates.  Unsurprisingly, in our setting it also seems a good choice.  Our recommendation is to set lambda=3 and not cross-validate this parameter.  We felt it was not a top line item to re-explore this issue in full depth in this paper.
> > >
> > > ## 3b.  Re: sensitivity of k
> > > We think that the algorithm is mostly invariant to k a very good aspect of the algorithm.
> > > As mentioned in the paper, we recommend k=d+2 as default.  This is because under some general position assumption, if k >= d+1 we obtain continuity, and this property becomes more stable as k increases.  On the other hand, as k increases, on a query the algorithm needs to access and evaluate and average more and more local functions.  This will make the algorithm inefficient, and less local.  This is why we did not explore this and why we did not feel it was an important direction to pursue.
> > >
> > >
> > > We hope this answers your concerns.  If not let us know.
> > > Thanks again for your engagement and feedback.

---

### Decision · Action_Editors · 2022-09-20

**Recommendation:** Accept with minor revision

**Comment:**

This paper proposes a new local kernel ridge regression method and analyzes its theoretical properties. The proposed method is simple yet effective, and the theoretical analysis of the adaptive KRR method is a good contribution. Moreover, the authors successfully address some of the concerns raised by the reviewers.

However, as reviewer j594 pointed out, the adaptive kernel width is relatively standard in practice; the main contribution is its theoretical analysis. Moreover, the hyperparameter sensitivity could be addressed more in the camera-ready version.  Therefore, we ask the authors to address the following issues in the camera-ready manuscript.

Main change:
1. The adaptive kernel width learning is a standard technique in practice, as pointed out by reviewer j594; claiming to propose an adaptive kernel method in this paper is too strong. As I wrote above, the key contribution would be the theoretical analysis of the adaptive kernel width method. So, please rewrite the introduction to emphasize the theoretical analysis of a Nadaraya-Watson-type method (not proposing adaptive KRR itself).

2. More careful evaluation of the hyperparameter selection raised by Reviewer j594. "The proposed method has a performance comparable to RBFInterpolator of scipy.interpolate, if we carefully choose several hyperparameters". In the revised paper, please clearly describe the hyper-parameter tuning procedure (90 percent of data is treated as training data, and 10 percent of data as test data?). Also, please run cross-validation or hold-out validation for the illustrative case and report the performance.

3. Related to Question 2. If the performance is comparable to scipy.interpolate in the illustrative example, please provide another example that the proposed method works, and discuss the reason carefully. If the performance of the proposed method is better than the global one in Question 2, the authors do not need to address this request.


Minor change:
1. Please make the font size of Figures 1 and 3 (left panels) bigger.
2. The font of Figure 2 is not consistent. Please use the same font for all figures.

---

> ### Author Response · Authors · 2022-09-29
> **Reply to Changes**
>
> We addressed the main changes in the camera-ready version submission as below:
>
> 1. Re: Adaptive NWKR
>
> In the abstract, we state the connection between Nadaraya-Watson type regression and our work by acknowledging that our method
>
> > can be interpreted as a variable-bandwidth Nadaraya-Watson Kernel Regression
>
> and further stress the theoretical analysis contribution of our work by saying:
>
> > the interpolated values provided by our local method can be proven to continuously vary with query points.
>
> At the start of Sec 4.1, we describe this connection in detail:
>
> > **Interpretation as NWKR estimator.** We first note that weighted interpolation of the local models can be interpreted as a variable-bandwidth Nadaraya-Watson kernel regression (NWKR) estimator with a triangle kernel $K(q,m) = \max\{0,1-\|m-q\|/h\}$ with bandwidth $h = r_k$ chosen as the distance to the $k$th nearest neighbor model.  Under NWKR, let $W = \sum_{m_i \in M} K(q,m_i)$ and the estimator at $q$ is $\frac{1}{W} \sum_{m_i \in M} w_i \mu_i(q)$.  With this triangle kernel, only the nearest $k-1$ models have non-zero weights, which are $K(q,m_i) = 1 - r_i/r_k$ and (via multiplying by $r_k/(r_k-r_1)$) are proportional to the weights considered in our interpolation $w_i = \frac{r_k - r_i}{r_k - r_1}$.
> While the continuity of fixed-bandwidth NWKR is known~\citep{ferraty2011kernel}, the following analysis can be interpreted as analyzing the continuity for this variable bandwidth condition under the triangle kernel.
>
>
>
> 2.  Hyper-parameter selection and RBFInterpolator
>
> For the 2D illustrative example, we now adopt the simulation setting proposed by reviewer j594, (the only change is previously we used 200 by 200 grid as test data, right now we use 201 by 201 grid).  We wrote:
>
> > This experiment also shows Simplex Method and python's RBF-Interpolator which does not have continuity guarantees. To tune hyper-parameters, we randomly select 10 percent of the data from training data as validation, then perform a grid-search over hyper-parameters space. We use the best-performed hyper-parameters on that validation set to evaluate the whole 201 by 201 grid.
>
> We report the RMSE and l-inf error in Table 1 for all the methods, and our local KRR performs best.
>
> |           | knn- svm | local- svm | local KRR | global KRR | RBF- Interpolator | simplex method |
> | --------- | ------- | --------- | ----------| -----| -------| -----------|
> | RMSE | 0.098 | 0.225 | 0.021 | 0.248 | 0.063 | 0.024 |
> | l_infty |  3.17 | 9.67 | 2.24 | 3.76 | 2.44 | 2.73 |
>
>
>
> 3.  Re: Sensitivity Analysis
>
> We provide a sensitivity analysis of hyper-parameters of our local KRR method in the Appendix. We wrote:
>
> > In that 10-fold cross-validation, each time we pick 10 percent of training data as the validation set, build the model based on the rest of the training data, and record the RMSE on the 10 percent predictions. We perform this procedure 10 times for random scenarios and report the average RMSE”.
> “We observe that there is very little variation with the choice of $k$ (default is $k=d+2$ and continuity requires $k \geq d+1$) and $\lambda$ (default is $ \lambda=3$, following literature).  However, $\ell$ is somewhat sensitive.  This is the main scale parameter that controls how many data points are required to measure a distinct patch of the model.  If $\ell$ is too small, the modeling in that patch may be too noisy, and if $\ell$ is too large, it may miss numerous small features.  This is a problem-dependent parameter -- balancing the noise vs. the size of features -- and so may not be surprising it benefits from some tuning.
>
> | k | RMSE |
> | - | ---- |
> | 2 | 0.035 |
> | 3 | 0.034 |
> | 4 | 0.031 |
> | 5 | 0.032 |
>
> | $\ell$ | RMSE |
> | ------ | ---- |
> | 10 | 0.055 |
> | 20 | 0.031 |
> | 30 | 0.056 |
> | 40 | 0.087 |
> | 50 | 0.134 |
>
> | $\lambda$ | RMSE |
> | --- | ---- |
> | 2 | 0.031 |
> | 3 | 0.031 |
> | 4 | 0.032 |
>
> 4.  Re: Minor requests
>
> We fixed the font issues in Figure 1,2,3.